# A Bayesian Nonparametric Perspective on Mahalanobis Distance for Out of Distribution Detection

**Randolph W. Linderman**                                    *randolph.linderman@duke.edu*
*Electrical and Computer Engineering Department*
*Duke University*

**Noah Cowan**                                               *ncowan@stanford.edu*
*Statistics Department*
*Stanford University*

**Yiran Chen**                                               *yiran.chen@duke.edu*
*Electrical and Computer Engineering Department*
*Duke University*

**Scott W. Linderman**                                       *scott.linderman@stanford.edu*
*Statistics Department and Wu Tsai Neurosciences Institute*
*Stanford University*

**Reviewed on OpenReview:** *https://openreview.net/forum?id=w3bMXPMDW1*

## Abstract

Bayesian nonparametric methods are naturally suited to the problem of out-of-distribution (OOD) detection. However, these techniques have largely been eschewed in favor of simpler methods based on distances between pre-trained or learned embeddings of data points. Here we show a formal relationship between Bayesian nonparametric models and the relative Mahalanobis distance score (RMDS), a commonly used method for OOD detection. Building on this connection, we propose Bayesian nonparametric mixture models with hierarchical priors that generalize the RMDS. We evaluate these models on the OpenOOD detection benchmark and show that Bayesian nonparametric methods can improve upon existing OOD methods, especially in regimes where training classes differ in their covariance structure and where there are relatively few data points per class.

## 1 Introduction

Machine learning systems inevitably face data that deviate from their training distributions. Generally, this data is either sparsely labeled or wholly unsupervised. Faced with such a dynamic environment, an intelligent system must accurately detect outliers and respond appropriately. This capability is the subject of modern research on out-of-distribution (OOD) detection (Hendrycks & Gimpel, 2017), anomaly detection Chandola et al. (2009), and open-set recognition (Scheirer et al., 2012).

Outlier detection is one of the oldest problems in statistics (Anscombe, 1960), and there are well-established methods for tackling this canonical problem. For example, Bayesian nonparametric methods offer a coherent, probabilistic framework for estimating the probability that a data point belongs to a new cluster (Ferguson, 1973; Antoniak, 1974; Lo, 1984; Sethuraman, 1994; MacEachern, 1994; Neal, 2000). Here, we use Dirichlet Process Mixture Models (DPMMs) to fit a generative model to the training data. Under this model, OOD detection reduces to a straightforward computation of the probability that a data point belongs to a novel class.

Care must be taken with Bayesian nonparametric methods, however. Modern machine learning systems are often built on foundation models that have been trained on massive datasets (Dosovitskiy et al., 2020;

Caron et al., 2021; Oquab et al., 2024; Darcet et al., 2024; Chen et al., 2020a;b). These models yield feature embeddings of data points that can be used for several downstream tasks, but the embeddings are high-dimensional. When fitting a Gaussian DPMM, for example, we must implicitly estimate the covariance of embeddings within and across classes, which presents both computational and statistical challenges. We propose a hierarchical model that adaptively shares statistical strength across classes when estimating these high-dimensional covariance matrices.

Generative classifiers like these have been largely eschewed in favor of simpler distance metrics, like the relative Mahalanobis distance score (RMDS; Ren et al., 2021). Here, we show both theoretically and empirically that RMDS is a close approximation to the outlier probability under a Gaussian DPMM with tied covariance matrices, connecting this widely-used approach to inference in a Bayesian nonparametric model. From this perspective, we propose hierarchical models that generalize RMDS by relaxing the assumption of equal covariance matrices across classes.

We find that hierarchical Gaussian DPMMs offer a well-grounded and practically competitive approach to OOD detection. Section 2 reviews related work, and Section 3 covers important background on DPMMs. We make a theoretical connection between RMDS and DPMMs in Section 4. This connection motivates our use of hierarchical models for estimating the high-dimensional covariance matrices in DPMMs — including a novel "coupled diagonal covariance" model — which we describe in Section 5 and evaluate in Section 6. We compare these models to existing baselines on synthetic datasets as well as the OpenOOD benchmark to characterize the regimes where Bayesian nonparameteric yield improved OOD performance [1].

## 2 Related Work

The OOD detection task for image classification is the binary decision that predicts if an image belongs to a class that is outside of the set of known classes that have been observed during training. This task has been widely studied and many solutions have been proposed. For example, some approaches alter the architecture or objective of a classifier (Tack et al., 2020; Huang & Li, 2021; Wei et al., 2022; Linderman et al., 2023), and others exploit auxiliary outlier datasets (Hendrycks et al., 2019; Zhang et al., 2023). While training and outlier exposure based methods are important research directions for OOD detection the focus of this work is post-hoc methods. We refer readers to the available high quality OOD detection surveys to learn more about these methods (Yang et al., 2022; Zhang et al., 2024). Our approach is related to a class of post-hoc methods including max softmax probability (MSP; Hendrycks & Gimpel, 2017), temperature-scaled MSP (Guo et al., 2017), ODIN (Liang et al., 2018), energy-based OOD Liu et al. (2020), the Mahalanobis distance score (MDS; Lee et al., 2018), and the Relative MDS (Ren et al., 2021), which derive OOD scores from embeddings or activations of a pre-trained network.

Recently, Zhang et al. (2024) proposed a set of Near and Far OOD benchmarks, as well as a leaderboard named OpenOOD to facilitate comparison across methods. The OpenOOD benchmarks found (1) that post-hoc methods are more scalable to large datasets, (2) there is no method that is best on all datasets, and (3) methods are sensitive to which model was used for embedding. The best performing OpenOOD methods for vision transformer (ViT) feature embeddings are the MDS and RMDS. The relative Mahalanobis distance score was inspired by earlier work by Ren et al. (2019) that addressed the poor performance of the OOD performance with density estimation methods. Sun et al. (2022) propose to relax some of the assumptions of Mahalanobis distance methods by using the negative $k$-th nearest neighbor distances instead. We will show that the relative Mahalanobis distance score (RMDS) is similar to scores derived from Bayesian nonparametric mixture models in Section 4.

Bayesian nonparametric methods have previously been proposed for outlier detection and used in several applications. Shotwell & Slate (2011) proposed to detect outliers within datasets by partitioning data via a DPMM and identifying clusters containing a small number of samples as outliers. Varadarajan et al. (2017) developed a method for detecting anomalous activity in video snippets by modeling object motion with DPMMs. Another line of work explored Dirichlet prior networks (DPN; Malinin & Gales, 2018; 2019) that explicitly model distributional uncertainty arising from dataset shift as a Dirichlet distribution over the

---

[1]The implementation for all models and experiments is available at `https://github.com/rw193/bnp4ood`.

categorical class probabilities. More recently, Kim et al. (2024) performed unsupervised anomaly detection through an ensemble of Gaussian DPMMs fit to random projections of a subset of datapoints. Our work focuses on connecting DPMMs to post-hoc confidence scores and developing *hierarchical* Gaussian DPMMs that share statistical strength across classes in order to estimate their high-dimensional covariance matrices.

## 3   Background

We start with background on Dirichlet process mixture models (DPMMs) and the special case of a Gaussian DPMM with tied covariance.

### 3.1   Dirichlet process mixture models

Dirichlet process mixture models (Lo, 1984) are Bayesian nonparametric models for clustering and density estimation that allow for a countably infinite number of clusters. There is always some probability that a new data point could come from a cluster that has never been seen before — i.e., that the new point is an *outlier*.

Let $\mathcal{D} = \{x_n, y_n\}_{n=1}^N$ denote a set of training data points $x_n \in \mathbb{R}^D$ and labels $y_n \in [K]$. Likewise, let $\mathcal{D}_k = \{x_n : y_n = k\}$ denote the subset of points assigned to cluster $k$, and let $N_k = |\mathcal{D}_k|$ denote the number of such points. Now consider a new, unlabeled data point $x$. Under a DPMM, its corresponding label $y$ has probability,

$$p(y = k \mid x, \mathcal{D}) \propto \begin{cases} N_k \, p(x \mid \mathcal{D}_k) & \text{if } k \in [K] \\ \alpha \, p(x) & \text{if } k = K + 1, \end{cases} \tag{1}$$

where the hyperparameter $\alpha \in \mathbb{R}_+$ specifies the concentration of the Dirichlet process prior.

The first case captures the probability that the new point belongs to one of the training clusters (that it is an *inlier*). That probability depends on two factors: 1) the number of training data points in that cluster since, intuitively, larger clusters are more likely; 2) the *posterior predictive probability*, which is obtained by integrating over the posterior distribution of cluster parameters, $\theta_k$,

$$p(x \mid \mathcal{D}_k) = \int p(x \mid \theta_k) \, p(\theta_k \mid \mathcal{D}_k) \, \mathrm{d}\theta_k \tag{2}$$

$$\propto \int p(x \mid \theta_k) \left[ \prod_{x_n \in \mathcal{D}_k} p(x_n \mid \theta_k) \right] p(\theta_k) \, \mathrm{d}\theta_k.$$

The second case of eq. (1) captures the probability that the new point is an outlier. It depends on the concentration $\alpha$ and the *prior predictive probability* obtained by integrating over the prior distribution of cluster parameters,

$$p(x) = \int p(x \mid \theta_k) \, p(\theta_k) \, \mathrm{d}\theta_k. \tag{3}$$

For many models of interest, the posterior and prior predictive distributions have closed forms.

### 3.2   Gaussian DPMM with Tied Covariance

For example, consider a Gaussian DPMM in which each cluster is parameterized by a mean and covariance, $\theta_k = (\mu_k, \Sigma_k)$. Assume a conjugate prior for the mean, $\mu_k \sim \mathcal{N}(\mu_0, \Sigma_0)$. For now, assume that all clusters share the same covariance matrix, which we express through an atomic prior, $\Sigma_k \sim \delta_\Sigma$. The hyperparameters of the prior are $\eta = (\mu_0, \Sigma_0, \Sigma)$.

Under this Gaussian DPMM, the conditional distribution of a new data point's label is,

$$p(y = k \mid x, \mathcal{D}) \propto \begin{cases} N_k \, \mathcal{N}(x \mid \mu_k', \Sigma_k' + \Sigma) & \text{if } k \in [K] \\ \alpha \, \mathcal{N}(x \mid \mu_0, \Sigma_0 + \Sigma) & \text{if } k = K + 1. \end{cases} \tag{4}$$

where

$$\mu'_k = \Sigma'_k \left( \Sigma_0^{-1} \mu_0 + N_k \Sigma^{-1} \bar{x}_k \right),$$
$$\Sigma'_k = \left( \Sigma_0^{-1} + N_k \Sigma^{-1} \right)^{-1}, \tag{5}$$

and $\bar{x}_k = \frac{1}{N_k} \sum_{x_n \in \mathcal{D}_k} x_n$ is the mean of the data points assigned to cluster $k$.

The relative probability of theses cases is an intuitive measure of how likely a point is to be an outlier. Indeed, the next section shows that the outlier probabilities from this Bayesian nonparametric model are closely related to another common outlier detection score.

## 4  Theory: Connecting Relative Mahalanobis Distance and DPMMs

Here we show that a widely used outlier detection method called the relative Mahalanobis distance score (RMDS; Ren et al., 2021) is closely related to the outlier probabilities obtained using a Gaussian DPMM with tied covariances. RMDS outputs a score, $C(x)$, where smaller values indicate that a data point $x$ is more likely to be an outlier. The RMDS score of a new point $x$ is defined as follows,[2]

$$\mathrm{MD}_0(x) = (x - \hat{\mu}_0)^\top \hat{\Sigma}_0^{-1} (x - \hat{\mu}_0)$$
$$\mathrm{MD}_k(x) = (x - \hat{\mu}_k)^\top \hat{\Sigma}^{-1} (x - \hat{\mu}_k)$$
$$\mathrm{RMD}_k(x) = \mathrm{MD}_0(x) - \mathrm{MD}_k(x)$$
$$C(x) = \max_k \ \mathrm{RMD}_k(x), \tag{6}$$

where $\mathrm{MD}_0(x)$ and $\mathrm{MD}_k(x)$ are squared Mahalanobis distances, $\hat{\mu}_0$ and $\hat{\Sigma}_0$ are the sample mean and covariance of the data, $\hat{\mu}_k$ is the sample mean of cluster $k$, and $\hat{\Sigma} = \frac{1}{N} \sum_{n=1}^N (x_n - \hat{\mu}_{y_n})(x_n - \hat{\mu}_{y_n})^\top$ is the sample within-class covariance.

Ren et al. (2021) motivated this score in terms of log density ratios between a Gaussian distribution for each cluster and a Gaussian "background" model. Specifically,

$$\mathrm{RMD}_k(x) = 2 \log \frac{\mathcal{N}(x \mid \hat{\mu}_k, \hat{\Sigma})}{\mathcal{N}(x \mid \hat{\mu}_0, \hat{\Sigma}_0)} + d, \tag{7}$$

where $d = \log |\hat{\Sigma}| - \log |\hat{\Sigma}_0|$ does not depend on $x$ or $k$. Larger values of $\mathrm{RMD}_k(x)$ indicate that $x$ is more likely under cluster $k$ than under the background model.

The procedure for mapping $\mathrm{RMD}_k(x)$ values to the score $C(x)$ is inherited from the Mahalanobis distance score (MDS; Lee et al., 2018). If the log density ratio is negative for all $k$, then the background model is more likely than *all* of the existing clusters, and hence $x$ is likely to be an outlier. Propositions 4.1 and 4.2 show that a similar computation is at work in the outlier probabilities for DPMMs.

First, we show that the *inlier* probabilities under a general DPMM (not necessarily Gaussian) can be expressed in terms of a quantity analogous to $C(x)$.

**Proposition 4.1.** *The* inlier *probability of a general DPMM with concentration $\alpha$ can be expressed as follows,*

$$p(y \in [K] \mid x, \mathcal{D}) = \sigma(\widetilde{C}(x) - \log \alpha / \overline{N}) \tag{8}$$

*where $\sigma(u) = (1 + e^{-u})^{-1}$ is the logistic function, $\overline{N} = \frac{1}{K} \sum_k N_k$ is the average cluster size, and*

$$\widetilde{C}(x) = \log \sum_{k=1}^K e^{\lambda_k + \log N_k / \overline{N}} \tag{9}$$

$$\lambda_k = \log \frac{p(x \mid \mathcal{D}_k)}{p(x)}. \tag{10}$$

*Here, $\lambda_k$ is the log density ratio of the posterior and prior predictive distributions from eq. (1).*

---

[2]We flip the sign of $\mathrm{RMD}_k(x)$ compared to Ren et al. (2021), but account for it in the definition of $C(x)$ so that the resulting score is unchanged. Our presentation is in keeping with the definition of the MDS score (Lee et al., 2018).

*Proof.* The inlier probability is one minus the outlier probability. Normalizing the outlier probability in eq. (1) and rearranging, we can write the inlier probability as,

$$p(y \in [K] \mid x, \mathcal{D}) = 1 - \frac{\alpha p(x)}{\alpha p(x) + \sum_{k=1}^{K} N_k p(x \mid \mathcal{D}_k)}$$

$$= 1 - \left( 1 + \sum_{k=1}^{K} \frac{N_k/\overline{N} \, p(x \mid \mathcal{D}_k)}{\alpha/\overline{N} \, p(x)} \right)^{-1}$$

$$= 1 - \left( 1 + e^{\widetilde{C}(x) - \log \alpha/\overline{N}} \right)^{-1}$$

$$= \sigma(\widetilde{C}(x) - \log \alpha/\overline{N}) \tag{11}$$

where $\widetilde{C}(x)$ is defined in eq. (9) and the last line follows from the fact that $1 - \sigma(-u) = \sigma(u)$. $\qquad \square$

This proposition says that the log-odds of data point $x$ belonging to an existing cluster is the difference of a *DPMM score*, $\widetilde{C}(x)$, which is analogous to the relative Mahalanobis score, and a *threshold*, $\log \alpha/\overline{N}$, which is tuned by the hyperparameter $\alpha$.

Next, we show that in certain regimes, the DPMM score from a Gaussian DPMM with tied covariance is almost perfectly correlated with the RMDS. Below, we define the relative covariance matrix $R = \hat{\Sigma}_0^{-1/2} \hat{\Sigma} \hat{\Sigma}_0^{-1/2}$ and note that as its operator norm $\kappa = \|R\|_{op}$ goes to 0, then we intuitively have that $\hat{\Sigma}_0$ is growing larger with respect to $\hat{\Sigma}$.

**Proposition 4.2.** *Consider $\sigma^2$-sub-Gaussian data in $\mathbb{R}^D$ generated from $K$ clusters with equal size $N$, where each cluster $k$ has mean $\mu_k$ and common covariance $\Sigma$. If the cluster means are drawn from a Gaussian prior $\mathcal{N}(\hat{\mu}_0, \hat{\Sigma}_0)$, then for any $\epsilon, \delta > 0$ there exist $\kappa_0, N_0$ such that if $\kappa \leq \kappa_0$ and $N \geq N_0$ then with probability at least $1 - \epsilon$,*

$$|\tilde{C}(X) - \tfrac{1}{2}[C(X) - d]| < \delta + \log K$$

*where $C(x)$ is the RMDS, $\tilde{C}(x)$ is the DPMM score from a Gaussian DPMM with tied covariance $\hat{\Sigma}$ and hyperparameters $(\hat{\mu}_0, \hat{\Sigma}_0, \hat{\Sigma})$, and $d$ is an additive constant, as defined in eq. (7).*

*Proof.* We give a sketch here and refer the reader to the full proof in Appendix E. For each $k$, we decompose the difference, $\lambda_k - \tfrac{1}{2}[\text{RMD}_k(x) - d]$, and show that it is small. The differences can be separated into terms that get smaller as $\kappa$ decreases,

$$\left| \log |\hat{\Sigma}_0 + \hat{\Sigma}| - \log |\hat{\Sigma}_0| \right|, \quad (x - \hat{\mu}_0)^T \left[ (\hat{\Sigma}_0 + \hat{\Sigma})^{-1} - \hat{\Sigma}_0^{-1} \right] (x - \hat{\mu}_0),$$

plus two more terms that decrease as $N$ increases,

$$\left| \log |\Sigma_k' + \hat{\Sigma}| - \log |\hat{\Sigma}| \right|, \quad (x - \hat{\mu}_k)^T \hat{\Sigma}^{-1} (x - \hat{\mu}_k) - (x - \mu_k')^T (\Sigma_k + \hat{\Sigma})^{-1} (x - \mu_k').$$

We collect these four terms and show that each is small with high probability. Finally, since the DPMM score is given by a log-sum-exp, which is 1-Lipschitz in the $\ell_\infty$ norm, it follows that if each $\lambda_k$ is within $\delta$ of $\tfrac{1}{2}[\text{RMD}_k(x) - d]$, then $\tilde{C}(x)$ is within $\delta + \log K$ of $\max_k \tfrac{1}{2}[\text{RMD}_k(x) - d] = \tfrac{1}{2}[C(x) - d]$. $\qquad \square$

This proposition establishes the close correspondence between the relative Mahalanobis distance score and the log-odds that a point is an inlier under a Gaussian DPMM with tied covariance. Note that the $\log K$ factor in Proposition 4.2 is irreducible due to the difference between the max used in RMDS and the smooth approximation used in DPMMs. We view this as a feature not a bug of the DPMMs: Since the $\log K$ factor only appears when the scores for different clusters are close to identical, the gap arises when the DPMM aggregates evidence across multiple equally plausible components rather than arbitrarily selecting a single cluster with RMDS. In practice, we find that a close correspondence holds in the experiments below and in Appendix I. This correspondence provides further support for using RMDS for outlier detection, beyond the original motivation in terms of log likelihood ratios. However, from this perspective, we also recognize several natural generalizations of RMDS that could improve outlier detection through richer DPMMs. We present three such generalizations below.

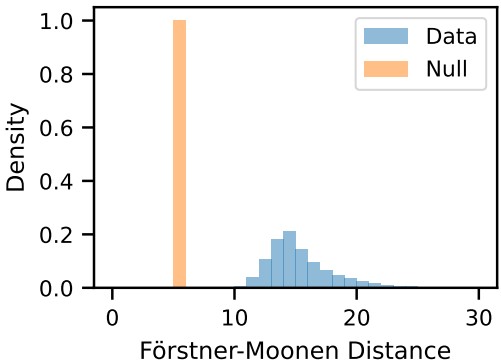

Figure 1: Förstner-Moonen distance between covariance matrices of the 1000 classes in the Imagenet-1k ViT-B-16 feature space (Data) versus 1000 samples of covariance matrices from the Wishart null distribution, $\text{W}(\overline{N}, \hat{\Sigma}/\overline{N})$. See Appendix B for complete details.

## 5 Hierarchical Gaussian DPMMs

RMDS has proven to be a highly effective outlier detection method, but it assumes that all clusters share the same covariance. This assumption helps avoid overfitting the covariance matrices for each class (Ren et al., 2021), but it is not always warranted. Figure 1 shows a histogram of differences between empirical covariance matrices $\hat{\Sigma}_k$ and $\hat{\Sigma}_{k'}$ for all pairs of classes $(k, k')$ in the Imagenet-1K dataset, as measured by the Förstner-Moonen distance (Förstner & Moonen, 2003).

These pairwise distances are systematically larger than what we would expect under a null distribution where the true covariance matrices are the same for all classes, and the empirical estimates differ solely due to sampling variability. Complete details of this analysis are provided in Appendix B. This analysis suggests that the covariance matrices are significantly different across classes and motivates a more flexible approach.

The connection between RMDS and Gaussian DPMMs established above suggests a natural way of relaxing the tied-covariance assumption without sacrificing statistical power: Instead of estimating covariance matrices independently, we could infer them jointly under a hierarchical Bayesian model (Gelman et al., 1995). With such a model, we can estimate separate covariance matrices for each cluster, while sharing information via a prior. By tuning the strength of the prior, we can obtain the tied covariance model in one limit and a fully independent model in the other. Finally, we can estimate these hierarchical prior parameters using a simple expectation-maximization algorithm that runs in a matter of minutes, even with large, high-dimensional datasets.

### 5.1 Full Covariance Model

First, we propose a hierarchical Gaussian DPMM with full covariance matrices and a conjugate prior. The cluster parameters, $\theta_k = (\mu_k, \Sigma_k)$, are drawn from a conjugate, normal-inverse Wishart (NIW) prior,

$$p(\theta_k) = \text{IW}\big(\Sigma_k \mid \nu_0, (\nu_0 - D - 1)\Sigma_0\big) \times \mathcal{N}\big(\mu_k \mid \mu_0, \kappa_0^{-1}\Sigma_k\big), \tag{12}$$

where IW denotes the inverse Wishart density. Under this parameterization, $\mathbb{E}[\Sigma_k] = \Sigma_0$ for $\nu_0 > D + 1$. The hyperparameters of the prior are $\eta = (\nu_0, \kappa_0, \mu_0, \Sigma_0)$.

The most important hyperparameters are $\nu_0$ and $\Sigma_0$, as they specify the prior on covariance matrices. As $\nu_0 \to \infty$, the prior concentrates around its mean and we recover a tied covariance model. For small values of $\nu_0$, the hierarchical model shares little strength across clusters, and the covariance estimates are effectively independent.

We propose a simple approach to estimate these hyperparameters in Appendix F. Briefly, we use empirical Bayes estimates for the prior mean and covariance, setting $\mu_0 = \hat{\mu}_0$ and $\Sigma_0 = \hat{\Sigma}$. We derive an expectation-

**A.**

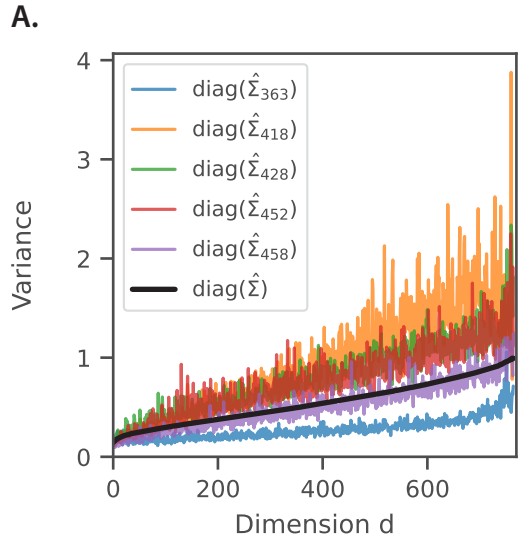

**B.**

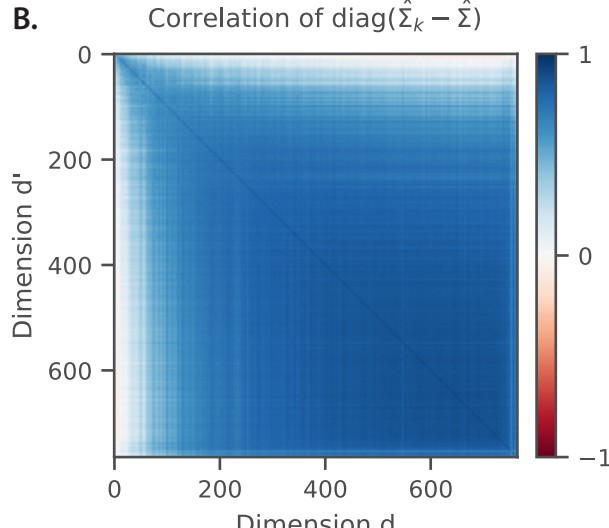

Figure 2: **A:** Diagonal of empirical covariance matrices, $\text{diag}(\hat{\Sigma}_k)$ for five randomly chosen clusters (colored lines) over dimensions. Compared to the diagonal of the average covariance matrix, $\text{diag}(\hat{\Sigma})$, individual clusters tend to have systematically larger or smaller variances than average. **B:** The correlation between dimensions of the deviation from the mean, $\hat{\Sigma}_k - \hat{\Sigma}$, of the diagonal components. The strong positive correlations between all but the first few dimensions indicates that the relationship observed in **A** is consistent across all clusters.

maximization (EM) algorithm to optimize $\nu_0$ and $\kappa_0$. Thanks to the conjugacy of the model, the E-step and the M-step for $\kappa_0$ can be computed in closed form. We leverage a generalized Newton method (Minka, 2000) to update the concentration hyperparameter, $\nu_0$, effectively learning the strength of the prior to maximize the marginal likelihood of the data.

Finally, the prior and posterior predictive distributions are multivariate Student's t distributions with closed-form densities. The log density ratios derived from these predictive distributions form the basis of the DPMM scores, $\widetilde{C}(x)$.

### 5.2 Diagonal Covariance Model

Even with the hierarchical prior, we find that the full covariance model can still overfit to high-dimensional embeddings. Thus, we also consider a simplified version of the hierarchical Gaussian DPMM with diagonal covariance matrices. Here, the cluser parameters are $\theta_k = \{\mu_{k,d}, \sigma_{k,d}^2\}_{d=1}^D$, and the conjugate prior is,

$$p(\theta_k) = \prod_{d=1}^D \chi^{-2}(\sigma_{k,d}^2 \mid \nu_{0,d}, \sigma_{0,d}^2) \times \mathcal{N}(\mu_{k,d} \mid \mu_{0,d}, \kappa_{0,d}^{-1}\sigma_{k,d}^2) \tag{13}$$

where $\chi^{-2}$ is the scaled inverse chi-squared density.

In addition to having fewer parameters per cluster, another advantage of this model is that it allows for different concentration hyperparameters for each dimension, $\nu_{0,d}$. We estimate the hyperparameters using a procedure that closely parallels the full covariance model. Likewise, the prior and posterior predictive densities reduce to products of scalar Student's t densities, which are even more efficient to compute. Complete details are in Appendix G.

### 5.3 Coupled Diagonal Covariance Model

The diagonal covariance model dramatically reduces the number of parameters per cluster, but it also makes a strong assumption about the per-class covariance matrices. Specifically, it assumes the variances, $\sigma_{k,d}^2$, are

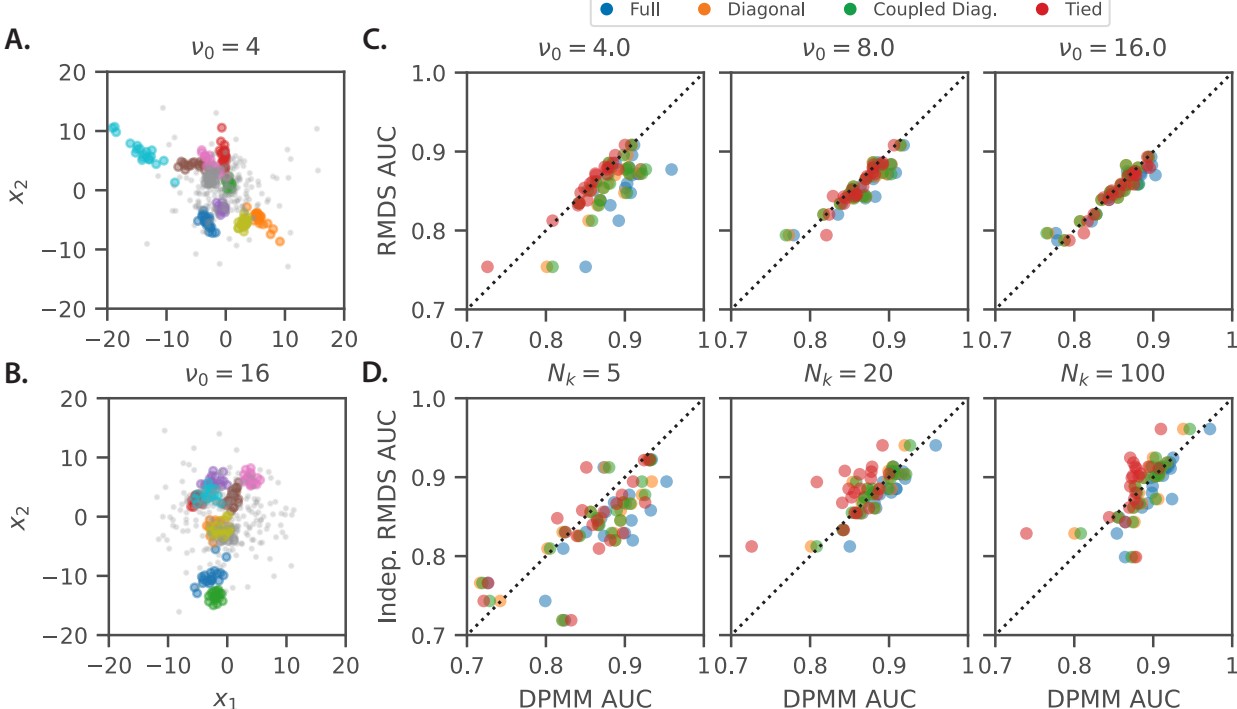

Figure 3: Synthetic experiments panel. Example sampled 2D dataset from DPMM with params $\nu_0 = 4$ (**A**) and 16 (**B**). Each data set has $K = 10$ clusters with $N_k = 20$ training data points each (colored dots). We evaluate performance on classifying outliers (gray dots) drawn from the prior predictive distribution. **C:** Performance of DPMM models vs. RMDS when sweeping over $\nu_0$ with $N_k = 20$ shows that DPMMs outperform when $\nu_0$ is small and there is greater variation in the $\Sigma_k$'s. **D:** Independent RMDS performance vs. DPMMs as a function of $N_k$ with $\nu_0 = 4$. Independent RMDS only performs well when there are adequate numbers of samples per class.

conditionally independent across dimensions. Figure 2 suggests that this is not the case: the diagonals of the empirical covariance matrices, $\hat{\Sigma}_k$, tend to be systematically larger or smaller than those of the average covariance matrix, $\hat{\Sigma}$. This analysis suggests that $\sigma^2_{k,d}$ are not independent; rather, if $\sigma^2_{k,d}$ is larger than average, then $\sigma^2_{k,d'}$ is likely to be larger as well.

We propose a novel, *coupled* diagonal covariance model to capture these effects. Specifically, we introduce a scale factor $\gamma_k \in \mathbb{R}_+$ that scales the variances for class $k$ compared to the average. In this model, the cluster parameters are $\theta_k = (\gamma_k, \{\mu_{k,d}, \sigma^2_{k,d}\}^D_{d=1})$, and the prior is,

$$p(\theta_k) = \chi^2(\gamma_k \mid \alpha_0) \prod_{d=1}^{D} \left[ \chi^{-2}(\sigma^2_{k,d} \mid \nu_{0,d}, \gamma_k \sigma^2_{0,d}) \times \mathcal{N}(\mu_{k,d} \mid \mu_{0,d}, \kappa^{-1}_{0,d} \sigma^2_{k,d}) \right] \tag{14}$$

where $\gamma_k$ scales the means of $\sigma^2_{k,d}$ for all dimensions $d$ to capture the correlations seen above.

Our procedure for hyperparameter estimation and computing DPMM scores is very similar to those described above. The only complication is that with the $\gamma_k$, the posterior distribution no longer has a simple closed form. However, for any fixed value of $\gamma_k$, the coupled model is a straightforward generalization of the diagonal model above. Since $\gamma_k$ is a one-dimensional variable, we can use numerical quadrature to integrate over its possible values. Likewise, we can estimate the hyperparameter $\alpha_0$ using a generalized Newton method, just like for the concentration parameter $\nu_0$. See Appendix H for complete details of this model.

# 6 Experiments

We experimentally tested these hierarchical Gaussian DPMMs on real and synthetic datasets. First we used simulated datasets to build intuition for where hierarchical models improve performance. Then we compared hierarchical Gaussian DPMMs to other widely-used OOD metrics on the OpenOOD Benchmark, and we studied performance versus dimensionality of the embeddings.

Table 1: Performance of Hierarchical Gaussian DPMM and baseline methods on the OpenOOD benchmark datasets Yang et al. (2022); Zhang et al. (2024), including 3 ID datasets (Imagenet-1K (Russakovsky et al., 2015) and CIFAR-10/100 (Krizhevsky et al., 2009)), and Near and Far OOD datasets. Accuracy of the classifiers on predicting the label $y \in [K]$ for in-distribution test data is reported for each benchmark. Other columns report AUROC scores for OOD detection on OpenOOD benchmark datasets. *We found the tied DPMM performance on Imagenet-1K improved when the prior parameter $\Sigma_0$ is set to the covariance of the data. For the CIFAR experiments we set $\Sigma_0$ to the covariance of the cluster means.

| | CIFAR-10 | | | CIFAR-100 | | | Imagenet-1K | | |
|---|---|---|---|---|---|---|---|---|---|
| Method | Accuracy | Near | Far | Accuracy | Near | Far | Accuracy | Near | Far |
| MSP | 94.93 | 88.36 | 91.80 | 76.19 | 80.33 | 78.88 | **80.90** | 75.80 | 86.30 |
| Temp. Scale | 94.93 | 88.43 | 91.84 | 76.19 | **80.57** | 79.25 | **80.90** | 77.29 | 88.62 |
| MDS | **95.04** | 85.41 | 90.15 | 76.10 | 58.86 | 69.29 | 80.41 | 78.97 | 92.57 |
| RMDS | **95.04** | 89.83 | 92.42 | 76.10 | 80.17 | 82.97 | 80.41 | 80.03 | 92.59 |
| | Hierarchical Gaussian DPMMs | | | | | | | | |
| Tied | **95.04** | 89.83 | 92.42 | 76.10 | 80.17 | **82.98** | 80.41* | 79.28* | **92.70***|
| Full | 94.95 | **90.63** | **93.50** | **76.64** | 79.22 | 81.20 | 76.78 | 70.66 | 86.31 |
| Diagonal | 94.76 | 89.14 | 90.86 | 76.07 | 79.22 | 82.88 | 76.54 | 80.60 | 90.85 |
| Coupled Diag. | 94.76 | 88.30 | 90.70 | 76.04 | 78.05 | 79.29 | 76.52 | **80.98** | 90.72 |

## 6.1 Synthetically generated dataset experiments

To understand the regimes in which hierarchical models yield benefits, we simulated $D = 2$ dimensional data from full covariance models with varying $\nu_0$ and $N_k$. When $\nu_0$ is small, covariances differ considerably across clusters, and the assumptions of the tied model (and of RMDS) are not well met. Conversely, as $\nu_0 \to \infty$, the prior concentrates on $\Sigma_0$, and the covariances are effectively tied. Figure 3A and 3B show simulated datasets from these two regimes. As expected, Figure 3C shows that hierarchical Gaussian DPMMs yielded considerable improvements when $\nu_0$ was small, and the largest improvements came from the full covariance model, which matched the data generating process. Notably, the tied model matched the RMDS performance, as predicted in Section 4.

We then asked if the hierarchical model was strictly necessary or whether a simpler model would suffice. For example, we considered an "Independent RMDS" based on Mahalanobis distances to the per-class covariance estimates, $\hat{\Sigma}_k$, instead of the average covariance $\hat{\Sigma}$. Intuitively, we expected the hierarchical models to perform best when there were few data points per cluster relative to the dimensionality of the embeddings; i.e., when $N_k/D$ is small. Indeed, Figure 3D shows that DPMMs offered substantial improvements in this regime, with diminishing gains as $N_k$ increased.

These analyses suggest that hierarchical Gaussian DPMMs should yield benefits in regimes where covariances differ across classes and the number of data points per class is relatively small.

## 6.2 OpenOOD Benchmark

Next, we compared hierarchical Gaussian DPMMs to other widely-used OOD detection methods on the OpenOOD benchmark datasets Yang et al. (2022); Zhang et al. (2024). The benchmark consists of 3 in-distribution (ID) datasets, CIFAR-10, CIFAR-100, and Imagenet-1K. For each ID dataset, several OOD datasets are grouped into *Near* and *Far*, where the Near OOD datasets are more similar to ID. For the

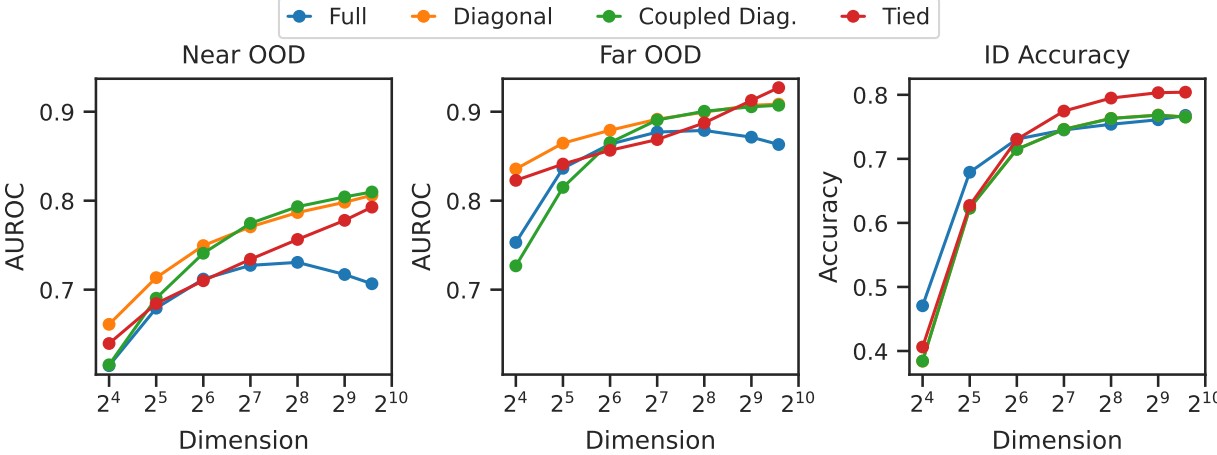

Figure 4: Performance on "near OOD", "far OOD", and in-distribution classification as a function of the feature dimension on the Imagenet-1K task. We projected the 768-dimensional ViT-B-16 features into lower dimensions using PCA, then projected into the eigenspace of the average within-class covariance. We compared the tied model (with full covariance) to the hierarchical model with full, diagonal, and coupled diagonal covariance and measured performance by area under the receiver operator curve (AUROC).

Imagenet-1K experiment, we used $D = 768$ dimensional embeddings from the ViT-B-16 model trained according to the DeIT method (Touvron et al., 2021), which are available in the Pytorch `torchvision` (TorchVision maintainers and contributors, 2016) package. The CIFAR experiments use OpenOOD's pretrained ResNet18 (He et al., 2016) features, $D = 512$. We preprocessed the embeddings as described in Appendix A. As baselines, we considered both MDS (Lee et al., 2018) and RMDS (Ren et al., 2021). We also compare to maximum softmax probability (MSP) (Hendrycks & Gimpel, 2017) and temperature scaled MSP with $T = 1000$ (Temp. Scale) (Guo et al., 2017) . We trained a single linear layer with gradient descent and supervised cross-entropy loss for the MSP methods. For all models, we measured the accuracy of classifying which class an in-distribution test image came from, as well as the AUROC score for outlier detection across the OpenOOD datasets. For DPMMs, we computed AUROC scores using $\widetilde{C}(x)$.

Table 1 shows that hierarchical models do offer improved performance, but the results are nuanced. We found the complexity and scale of the ID dataset determines which modeling assumptions are most appropriate. The full covariance model yields the highest performance on Near and Far OOD on the small scale CIFAR-10 task. Whereas the coupled diagonal model is the best performing model in Near OOD settings for the large-scale Imagenet-1K task. The full covariance model performed surprisingly poorly on the Imagenet-1K task, and we suspected it was due to the high-dimensional embeddings. We provide expanded results tables for each ID dataset that show performance on each OOD dataset in Appendices J, K and M.

## 6.3 Performance vs. Dimensionality

Finally, we asked how these methods compare as we vary the dimensionality of the embeddings. We suspected that the full covariance model would perform better in lower dimensions for two reasons. First, it has $\mathcal{O}(KD^2)$ parameters compared to only $\mathcal{O}(KD)$ in the diagonal models and $\mathcal{O}(D^2)$ in the tied model, so even with a hierarchical prior, the full model could still overfit. This problem is exacerbated for classes that have fewer data points than the feature dimension, in which case the prior has a strong effect on the conditional distribution of the per-class covariance matrix and the posterior predictive distributions. Second, we suspected that the inverse Wishart prior distribution, which has only a scalar concentration $\nu_0$, may be a poor prior for high-dimensional covariance matrices.

To test this hypothesis, we swept the number of principal components retained in preprocessing (see Appendix A). We found that out-of-distribution detection of the full-covariance hierarchical model plateaued

for $D \geq 128$ dimensional embeddings (Figure 4). By contrast, the diagonal and coupled diagonal models performed considerably better, especially on Near OOD benchmarks. The diagonal covariance model outperforms the tied model across all dimensions on Near OOD detection, as well as in lower dimensions for Far OOD. However, in-distribution classification accuracy plateaus around 256 dimensions.

Altogether, these analyses of synthetic and real datasets show that hierarchical Gaussian DPMMs are advantageous for OOD detection, especially in regimes where: *i)* covariance matrices differ across clusters; *ii)* the number of data points per cluster is small compared to the dimension; and *iii)* detection relies on fine-grained distinctions between training data and Near OOD test points.

## 7    Discussion

We developed a theoretical connection between the relative Mahalanobis distance score for outlier detection and the outlier probability under a Gaussian DPMM with tied covariance. This Bayesian nonparametric perpsective led us to propose hierarchical Gaussian DPMMs that allow each cluster a different covariance matrix, while still sharing statistical strength across classes via the prior. We developed efficient EM algorithms to estimate the hyperparameters of the hierarchical models, and we studied their performance on synthetic data as well as the OpenOOD benchmarks. We found that these models — especially the coupled diagonal covariance model — yielded improved performance on some benchmarks, especially the Near OOD benchmarks.

**Limitations and Future Work**    Despite the hierarchical prior, we found that the full covariance Gaussian DPMMs were prone to overfitting. Further work could explore low-rank plus diagonal covariance matrices, which would interpolate between the diagonal and full covariance models. More generally, like RMDS and MDS, we assume that features are Gaussian distributed within each class. The competitive performance on the OpenOOD Imagenet benchmark using ViT features suggests that this assumption is reasonable (Yang et al., 2022; Zhang et al., 2024), but there is no guarantee. Future work could consider fine-tuning the features or learning a nonlinear transformation to address this potential source of model misspecification, as in prototype networks (Snell et al., 2017).

Bayesian nonparametric approaches naturally extend to other closely related problems, like generalized category discovery Vaze et al. (2022a) and continual learning Van de Ven et al. (2022). For example, given a collection of data points, a DPMM may have sufficient evidence to allocate new classes for the out-of-distribution data. More generally, casting OOD as inference in a generative model brings modeling choices to the fore. Here, we focused on the challenge of modeling high-dimensional covariance matrices that may vary across classes, but there are several other ways in which the simple Gaussian DPMM could be improved. For example, we could attempt to capture the nonstationarity inherent in the OOD setting by allowing the prior predictive distribution to drift from what was inferred based on the training data. Such a model could afford greater robustness on OOD detection tasks.

**Conclusion**    In summary, we find that Bayesian nonparametric methods with hierarchical priors are a promising approach for OOD detection. If the features extracted from foundation models are reasonably well approximated as realizations of Gaussian DPMMs, the posterior inferences under such models can provide accurate estimates of outlier probability. This probabilistic perspective not only casts widely used methods in a new light, it also leads to practical model improvements and enables several lines of future inquiry.

### Acknowledgments

We thank Jingyang Zhang for his feedback and detailed knowledge of the OpenOOD codebase. RWL was supported in part by the National Science Foundation (NSF) under Grant No. 2112562 to YC, the U.S. Army Research Laboratory and the U.S. Army Research Office (ARL/ARO) under grant number ARO-W911NF-23-2-0224 to YC, and the Department of Defense (DoD) through the National Defense Science & Engineering Graduate (NDSEG) Fellowship Program. SWL was supported in part by fellowships from the Simons Collaboration on the Global Brain, the Alfred P. Sloan Foundation, and the McKnight Foundation.

Any opinions, findings, and conclusions or recommendations expressed in this material are those of the authors and do not necessarily reflect the views of the NSF, the ARL/ARO, or the DoD.

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

## A  Preprocessing

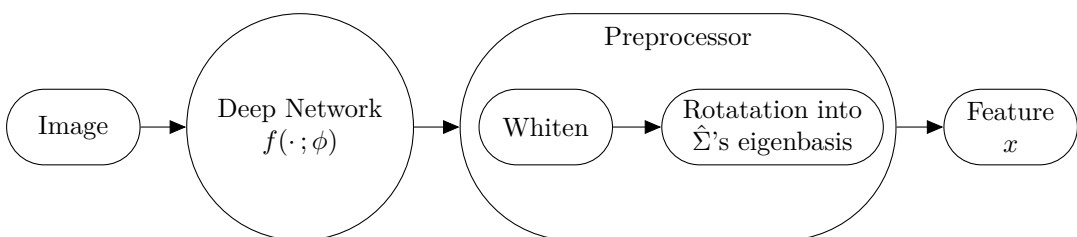

Figure 5: W&R preprocessing flowchart.

Before computing OOD scores, we first preprocessed the embeddings using PCA to whiten and sort the dimensions in order of decreasing variance. We discarded dimensions with near zero variance to ensure the empirical covariance matrices were full rank. We scaled each dimension by the inverse square root of the eigenvalues so that the transformed embeddings had identity covariance. Finally, we rotated the embeddings using the eigenvectors of the average covariance matrix, so that the average within-class covariance matrix is diagonal.

More precisely, the preprocessing steps are as follows:

1. Let $\{x_i\}_{i=1}^N$ denote the mean-centered embeddings.

2. Let $\hat{\Sigma}_0 = U \Lambda U^\top$ denote the covariance of the centered embeddings and its eigendecomposition. Discard any dimensions with eigenvalues less than a threshold of approximately $10^{-7}$, and then we project and scale the embeddings by,

$$x_i' \leftarrow \Lambda^{-\frac{1}{2}} U^T x_i, \tag{15}$$

so that the empirical covariance of $\{x_i'\}_{i=1}^n$ is the identity matrix.

3. Compute the average within-class covariance $\hat{\Sigma} = \frac{1}{N} \sum_{i=1}^M (x_i' - \hat{\mu}_{y_i})(x_i' - \hat{\mu}_{y_i})^\top$, where $\hat{\mu}_k = \frac{1}{N_k} \sum_{i:y_i=k} x_i'$.

4. Compute the eigendecomposition $\hat{\Sigma} = V S V^\top$, with eigenvalues $S = \mathrm{diag}(\sigma_1^2, \ldots, \sigma_D^2)$ sorted in increasing order of magnitude so that the first dimension has the smallest within-class covariance. The embeddings $x_i'$ have unit variance in all dimensions, but along the dimension of the first eigenvector in $V$, the average within-class covariance is smallest.

5. Project the embeddings into this eigenbasis,

$$z_i \leftarrow V^\top x_i'. \tag{16}$$

After these preprocessing steps, the resulting embeddings $\{z_i\}_{i=1}^N$ are zero mean ($\hat{\mu}_0 = 0$), their empirical covariance is the identity ($\hat{\Sigma}_0 = I$), and the average within-class covariance is diagonal ($\hat{\Sigma} = \mathrm{diag}(\sigma_1^2, \ldots, \sigma_D^2)$). The empirical within-class covariance matrix $\hat{\Sigma}_k$ for class $k$ will *not* generally be diagonal, but this sequence of preprocessing steps is intended to make them closer to diagonal on average.

Note that the relative Mahalanobis distance score is invariant to these linear transformations. They simply render the embeddings more amenable to our hierarchical models with diagonal covariance. We further test the effect of these preprocessing steps via the ablation experiment described in Appendix J.

## B  Further Details of Exploratory Analyses

First, we investigated the degree to which the sample covariance matrices differ between the 1000 classes in the Imagenet-1K dataset. In accordance with the OpenOOD benchmark, we used embeddings from the ViT-B-16 model trained according to the DeIT method (Touvron et al., 2021), which are $D = 768$ dimensional.

The embeddings are available in the Pytorch `torchvision` (TorchVision maintainers and contributors, 2016) package. We preprocessed the embeddings as described in Appendix A. For this analysis, we only kept the top 128 PCs to speed computation.

To measure the distance between covariance matrices for two clusters, we used the Förstner-Moonen (FM) metric (Förstner & Moonen, 2003),

$$d(\boldsymbol{\Sigma}_1, \boldsymbol{\Sigma}_2) = \left[ \sum_{i=1}^{n} (\log \lambda_i(\boldsymbol{\Sigma}_1^{-1} \boldsymbol{\Sigma}_2))^2 \right]^{\frac{1}{2}}, \tag{17}$$

where $\lambda_i(\boldsymbol{\Sigma}_1^{-1} \boldsymbol{\Sigma}_2)$ is the $i$-th eigenvalue of $\boldsymbol{\Sigma}_1^{-1} \boldsymbol{\Sigma}_2$. We computed the FM metric for all pair of empirical covariance matrices $(\hat{\Sigma}_k, \hat{\Sigma}_{k'})$. We compared the distribution of FM distances under the real data to distances between covariance matrices sampled from the null model, in which $\Sigma_k$ truly equals $\hat{\Sigma}$ for all $k$, and the differences in the estimates $\hat{\Sigma}_k$ arise solely from sampling error. The corresponding null distribution is a Wishart distribution, $\hat{\Sigma}_k \sim \mathrm{W}(\overline{N}, \hat{\Sigma}/\overline{N})$, where $\overline{N}$ is the average number of data points per class.

## C  Compute Resources

The OOD experiments were performed on a cluster consisting of compute nodes with 8 NVIDIA RTX A5000 GPUs. The OpenOOD (Yang et al., 2022; Zhang et al., 2024) experiments on the Imagenet-1K dataset (Russakovsky et al., 2015) utilized weights available from Pytorch's (Paszke et al., 2019) torchvision (TorchVision maintainers and contributors, 2016) models. The compute across experiments was reduced by storing summary statistics of the embeddings for the Gaussian models. The DPMM fitting and prediction was performed on the CPU except for calculating the posterior and prior predictive distributions for each sample.

## D  Computational Complexity

The complexity of computing outlier scores is not that much greater than computing the relative Mahalanobis distance score (RMDS). Our preprocessing procedure (Appendix A) involves whitening and rotating the embeddings, which requires $\mathcal{O}(D^2)$ memory and $\mathcal{O}(D^3)$ time, just like RMDS.

In order to account for different covariances across classes, we incur additional costs. The first costs are those of estimating the hyperparameters via expectation-maximization (EM). In the EM algorithm, we need to store expected covariance matrices (or rather, their inverses) for each class. This costs $O(KD^2)$ space for the full covariance model and $\mathcal{O}(KD)$ space for the (coupled) diagonal model. Each M-step of the EM algorithm also involves inverting covariance matrices, so for the full covariance model the time complexity is $\mathcal{O}(TKD^3)$, and for the diagonal and coupled-diagonal covariance model it is only $\mathcal{O}(TKD)$, where $T$ is the number of EM iterations. Typically, we find that $T \approx 10$ iterations is enough to converge.

Finally, there is the cost of computing the final outlier scores, which involves computing Mahalanobis distances for RMDS and a marginal likelihood ratio the DPMM. For both RMDS and full covariance DPMM scores, these can be computed in $\mathcal{O}(ND^2)$ time using the Cholesky factors of the covariance matrices from the preprocessing or EM algorithm; for the (coupled) diagonal covariance model, this cost is only $\mathcal{O}(ND)$.

In sum, the additional expressivity of the Bayesian nonparametric mixture models incurs some additional cost. For the full covariance model, these costs can become burdensome (especially in memory), but for the diagonal and coupled diagonal model they are quite reasonable and, as we show in our experiments, they can lend a significant boost.

## E  Proof of Proposition 4.2

Using the definitions, we can write $\lambda_k$ exactly as a difference of log Gaussian densities:

$$\lambda_k = \log \frac{\mathcal{N}(x \mid \mu'_k, \Sigma'_k + \hat{\Sigma})}{\mathcal{N}(x \mid \hat{\mu}_0, \hat{\Sigma}_0 + \hat{\Sigma})} = \left[ \log \mathcal{N}(x \mid \mu'_k, \Sigma'_k + \hat{\Sigma}) - \log \mathcal{N}(x \mid \hat{\mu}_0, \hat{\Sigma}_0 + \hat{\Sigma}) \right].$$

Similarly, the RMDS for cluster $k$ can be written as a log-density ratio between a cluster Gaussian and the "background" Gaussian model:

$$\text{RMD}_k(x) \;=\; 2\log \frac{\mathcal{N}(x \mid \hat{\mu}_k, \hat{\Sigma})}{\mathcal{N}(x \mid \hat{\mu}_0, \hat{\Sigma}_0)} + d,$$

where $d = \log|\hat{\Sigma}| - \log|\hat{\Sigma}_0|$. For clarity, define the ideal log-ratio using true cluster parameters as

$$\rho_k(x) \;:=\; \log \frac{\mathcal{N}(x \mid \mu_k, \Sigma)}{\mathcal{N}(x \mid \mu_0, \Sigma_0)}.$$

Notice that $\frac{1}{2}[\text{RMD}_k(x) - d] = \log \frac{\mathcal{N}(x \mid \hat{\mu}_k, \hat{\Sigma})}{\mathcal{N}(x \mid \hat{\mu}_0, \hat{\Sigma}_0)} =: \hat{\rho}_k(x)$ is the same ratio but with empirical estimates $(\hat{\mu}_k, \hat{\Sigma})$ in place of $(\mu_k, \Sigma)$, and $\rho_k(x)$ uses the true cluster mean and covariance. We wish to bound $2|\lambda_k - \hat{\rho}_k|$. We now can use the triangle inequality and decompose this into four separate terms:

$$2|\lambda_k - \hat{\rho}_k| \le \left| \log|\hat{\Sigma}_0 + \hat{\Sigma}| - \log|\hat{\Sigma}_0| \right| \Big\} \tag{A}$$

$$+ \left| (x - \hat{\mu}_0)^T \left[ (\hat{\Sigma}_0 + \hat{\Sigma})^{-1} - \hat{\Sigma}_0^{-1} \right] (x - \hat{\mu}_0) \right| \Big\} \tag{B}$$

$$+ \left| \log|\Sigma'_k + \hat{\Sigma}| - \log|\hat{\Sigma}| \right| \Big\} \tag{C}$$

$$+ \left| (x - \hat{\mu}_k)^T \hat{\Sigma}^{-1}(x - \hat{\mu}_k) - (x - \mu'_k)^T (\Sigma_k + \hat{\Sigma})^{-1}(x - \mu'_k) \right| \Big\} \tag{D}$$

To bound A, we first note that

$$\log|\hat{\Sigma}_0 + \hat{\Sigma}| - \log|\hat{\Sigma}_0| = \log|I + \hat{\Sigma}_0^{-1}\hat{\Sigma}| = \log|I + \hat{\Sigma}_0^{-1/2}\hat{\Sigma}\hat{\Sigma}_0^{-1/2}| = \log|I + R|$$

using the fact that the determinant behaves multiplicatively with respect to the matrix product. Then given that the determinant is the product of the eigenvalues we finish our bound on A

$$\text{A} \le \sum_{i=1}^{D} \log(1 + \lambda_i^{(R)}) \le D\log(1 + \kappa) \le D\kappa$$

thus this term is less than $\delta_1$ when $\kappa < \kappa_1 \le \frac{\delta_1}{D}$. For term B, recall that

$$(A + B)^{-1} = A^{-1} - A^{-1}B(A + B)^{-1}$$

(which can be verified by right multiplying by $A + B$) combined with the fact that

$$\begin{aligned}
\hat{\Sigma}_0^{1/2}\hat{\Sigma}_0^{-1}\hat{\Sigma}(\hat{\Sigma}_0 + \hat{\Sigma})^{-1}\hat{\Sigma}_0^{1/2} &= \hat{\Sigma}_0^{-1/2}\hat{\Sigma}(\hat{\Sigma}_0 + \hat{\Sigma})^{-1}\hat{\Sigma}_0^{1/2} \\
&= \hat{\Sigma}_0^{-1/2}\hat{\Sigma}(\hat{\Sigma}_0(I + \hat{\Sigma}_0^{-1}\hat{\Sigma}))^{-1}\hat{\Sigma}_0^{1/2} \\
&= \hat{\Sigma}_0^{-1/2}\hat{\Sigma}(I + \hat{\Sigma}_0^{-1}\hat{\Sigma})^{-1}\hat{\Sigma}_0^{-1}\hat{\Sigma}_0^{1/2} \\
&= \hat{\Sigma}_0^{-1/2}\hat{\Sigma}(I + \hat{\Sigma}_0^{-1}\hat{\Sigma})^{-1}\hat{\Sigma}_0^{-1/2} \\
&= \hat{\Sigma}_0^{-1/2}\hat{\Sigma}[\hat{\Sigma}_0^{-1/2}(I + R)\hat{\Sigma}_0^{1/2}]^{-1}\hat{\Sigma}_0^{-1/2} \\
&= R[1 + R]^{-1}
\end{aligned}$$

Means that we can say

$$\hat{\Sigma}_0^{-1}\hat{\Sigma}(\hat{\Sigma}_0 + \hat{\Sigma})^{-1} \preceq \frac{\kappa}{1 + \kappa}\hat{\Sigma}_0^{-1}$$

By the standard fact that for $B$ positive definite, $A^{-1/2}BA^{-1/2} \preceq \alpha I$ if and only if $B \preceq \alpha A$. So combining all of these identities we have

$$\mathsf{B} \le \frac{\kappa}{1+\kappa}(x - \hat\mu_0)^\top \hat\Sigma_0^{-1}(x - \hat\mu_0)$$

Which we then bound with the standard Hanson Wright inequality Vershynin (2018) to get that with probability $1 - \epsilon_1$

$$\mathsf{B} \le \kappa\sigma^2 (D + \sqrt{D\log\frac{1}{\epsilon_1}} + 2\log\frac{1}{\epsilon_1})$$

Which is less than $\delta_2$ when $\kappa < \kappa_2 = \left[\sigma^2(D + \sqrt{D\log\frac{1}{\epsilon_1}} + 2\log\frac{1}{\epsilon_1})\right]^{-1}\delta_2$

Now to deal with term $\mathsf{C}$, we again note the multiplicativity of the determinant with respect to matrix products and see that

$$\log|\Sigma'_k + \hat\Sigma| - \log|\hat\Sigma| = \log|I + \hat\Sigma^{-1}\Sigma'_k|$$

But recall that $\Sigma'_k = (\hat\Sigma_0^{-1} + N\hat\Sigma^{-1})^{-1}$, which immediately gives a useful result when combined with the fact that $\hat\Sigma_0^{-1}\hat\Sigma = \hat\Sigma_0^{-1/2}R\hat\Sigma_0^{1/2}$

$$
\begin{aligned}
\log|I + \hat\Sigma^{-1}\Sigma'_k| &= \log|I + \hat\Sigma^{-1}(\hat\Sigma_0^{-1} + N\hat\Sigma^{-1})^{-1}| \\
&= \log|I + (NI + \hat\Sigma_0^{-1}\hat\Sigma)^{-1}| \\
&= \log|I + (NI + \hat\Sigma_0^{-1/2}R\hat\Sigma_0^{1/2})^{-1}| \\
&= \log|\hat\Sigma_0^{1/2}\hat\Sigma_0^{-1/2} + (N\hat\Sigma_0^{-1/2}\hat\Sigma_0^{1/2} + \hat\Sigma_0^{-1/2}R\hat\Sigma_0^{1/2})^{-1}| \\
&= \log|\hat\Sigma_0^{1/2}\hat\Sigma_0^{-1/2} + \hat\Sigma_0^{1/2}(NI + R)^{-1}\hat\Sigma_0^{-1/2}| \\
&= \log|\hat\Sigma_0^{1/2}||I + (NI + R)^{-1}||\hat\Sigma_0^{-1/2}| \\
&= \log|I + (NI + R)^{-1}|
\end{aligned}
$$

Now we are functionally done, since we can bound this easily using known bounds of the logarithm (easy to check by concavity) to say

$$\log|I + (NI + R)^{-1}| = \sum_{i=1}^{D}\log\left(1 + \frac{1}{N + \lambda_i}\right)$$

$$\mathsf{C} \le D\log\left(1 + \frac{1}{N}\right) \le \frac{D}{N}$$

So when $N \ge N_1 > \frac{D}{\delta_3}$ then $\mathsf{C} \le \delta_3$.

Finally, we turn to $\mathsf{D}$. We note that we can re-write it as

$$(x - \mu'_k + \mu'_k - \hat\mu_k)^T\hat\Sigma^{-1}(x - \mu'_k + \mu'_k - \hat\mu_k) - (x - \mu'_k)^T(\Sigma_k + \hat\Sigma)^{-1}(x - \mu'_k)$$

which decomposes into three parts. Trying to ease notation, let $v = x - \mu'_k$ and $u = \mu'_k - \hat\mu_k$. So then $\mathsf{D}$ is just

$$\underbrace{v^\top[\hat\Sigma^{-1} - (\Sigma_k + \hat\Sigma)^{-1}]v}_{(i)} + \underbrace{u^\top\hat\Sigma^{-1}u}_{(ii)} + \underbrace{2u^\top\hat\Sigma^{-1}v}_{(iii)}$$

For the first piece, we again use the identity

$$(A + B)^{-1} = A^{-1} - A^{-1}B(A + B)^{-1}$$

to get that $(i)$ is equal to

$$v^\top\hat\Sigma^{-1}\Sigma'_k(\hat\Sigma + \Sigma'_k)^{-1}v$$

And recall the Loewner-Heinz inequality Löwner (1934); Carlen (2010) tells us that when $A + B \succ A$, then $(A + B)^{-1} \prec A^{-1}$. Then note that we can show with not too much difficulty (using the fact that since $A$ and $B$ are positive definite then $B^{-1/2} A B^{-1/2} \prec I \iff A \prec B$) that this implies

$$\hat{\Sigma}^{-1} \Sigma'_k (\hat{\Sigma} + \Sigma'_k)^{-1} \preceq \hat{\Sigma}^{-1} \Sigma'_k \hat{\Sigma}^{-1}$$

But since $\Sigma'_k \preceq \frac{1}{N} \hat{\Sigma}$ we get that $(i)$ is bounded by

$$\frac{1}{N} v^\top \hat{\Sigma}^{-1} v$$

now here we again use Hanson Wright to say with probability at least $1 - \epsilon_2$, as long as $N > N_2 = \frac{1}{\delta_4} \sigma^2 \left( D + \sqrt{D \log \frac{1}{\epsilon_2}} + 2 \log \frac{1}{\epsilon_2} \right)$ it is bounded. For term $(ii)$, we have rapid convergence of $\mu'_k$ to $\hat{\mu}_k$ and note that $\hat{\Sigma}^{-1}$ is bounded above by $\frac{1}{N} \|\hat{\Sigma}\|_{op}$. While the difference between the means is bounded by $\frac{1}{N} \left( 2\|\hat{\Sigma}\|_{op} \lambda_{\min}(\hat{\Sigma}_0) \right)$ so if $N \geq N_3 = \frac{1}{\delta_5} \left( 2\|\hat{\Sigma}\|_{op} \lambda_{\min}(\hat{\Sigma}_0) \right)$, then we have convergence. Finally, we can just use Cauchy-Schwarz to argue that if $N$ is set as above, then the bounds for both of the above hold for the cross term. Combining all terms together, we have that if $\kappa < \min\{\kappa_1, \kappa_2\}$ and $N > \max\{N_1, N_2, N_3\}$, then the difference between $\lambda_k$ and $\rho_k$ is less than $\delta = \delta_1 + \delta_2 + \delta_3 + \delta_4 + \delta_5$ with probability $1 - \epsilon$ as long as $\epsilon_1 + \epsilon_2 \leq \epsilon$.

Finally,

$$\tilde{C}(x) = \log \sum_k \exp\big(\lambda_k(x)\big).$$

The map $(\lambda_1, \ldots, \lambda_K) \mapsto \log\big(\sum_k e^{\lambda_k}\big)$ is 1-Lipschitz continuous in the $\ell_\infty$-norm in the sense that if $|\lambda_k - \lambda'_k| \leq \delta$ for all $k$, then $\big|\text{LogSumExp}\{\lambda_k\} - \text{LogSumExp}\{\lambda'_k\}\big| \leq \epsilon$. Hence if $\lambda_k(x)$ is within $\delta$ of $\frac{1}{2}\left[\text{RMD}_k(x) - d\right]$ uniformly in $k$, then $\tilde{C}(x)$ is within $\delta$ of $\text{LogSumExp}_k\{\frac{1}{2}\text{RMD}_k(x) - d\}$. Noting that

$$|\text{LogSumExp}\{\lambda_k\} - \max_k\{C_k(x)\}| \leq |\text{LogSumExp}\{\lambda_k\} - \text{LogSumExp}\{C_k(x)\}|$$

$$+ |\text{LogSumExp}\{C_k(x)\} - \max_k C_k(x)|$$

$$\leq \delta + \log K$$

Thus we conclude

$$\big| \tilde{C}(x) - \left[\tfrac{1}{2} C(x) - d\right] \big| \leq \delta + \log K,$$

with probability at least $1 - \epsilon$. This completes the proof.

# F   EM Algorithm for the Full Covariance Model

Here we describe an expectation-maximization (EM) algorithm for estimating the hyperparameters of the hierarchical covariance model. Recall that under this model,

$$\Sigma_k \sim \text{IW}(\nu_0, (\nu_0 - D - 1)\Sigma_0)$$
$$\mu_k \sim \text{N}(\mu_0, \kappa_0^{-1}\Sigma_k)$$

so that the prior hyperparameter $\Sigma_0$ specifies the mean of the per-class covariances, $\mathbb{E}[\Sigma_k] = \Sigma_0$. This hyperparameter should not be confused with $\hat{\Sigma}_0$ defined in the main text, which denoted the empirical estimate of the marginal covariance. Also note that this prior formulation requires $\nu_0 > D + 1$.

First, we set the hyperparameters $\mu_0$ and $\Sigma_0$ using empirical Bayes estimates,

$$\mu_0 = \hat{\mu}_0 = \frac{1}{N} \sum_{n=1}^{N} x_n \tag{18}$$

$$\Sigma_0 = \hat{\Sigma} = \frac{1}{N} \sum_{n=1}^{N} (x_n - \hat{\mu}_{y_n})(x_n - \hat{\mu}_{y_n})^\top \tag{19}$$

where $\hat{\mu}_{y_n} = \frac{1}{N_k} \sum_{n:y_n=k} x_n$.

To set the remaining hyperparameters, $\kappa_0$ and $\nu_0$, we use EM. The expected log likelihood is separable over these two hyperparameters,

$$\mathcal{L}(\nu_0, \kappa_0) = \mathcal{L}(\nu_0) + \mathcal{L}(\kappa_0)$$

$$\mathcal{L}(\nu_0) = \mathbb{E}\left[\sum_{k=1}^{K} \log \text{IW}(\Sigma_k \mid \nu_0, (\nu_0 - D - 1)\Sigma_0)\right]$$

$$\mathcal{L}(\kappa_0) = \mathbb{E}\left[\sum_{k=1}^{K} \log \text{N}(\mu_k \mid \mu_0, \kappa_0^{-1}\Sigma_k)\right]$$

where the expectations are taken with respect to the posterior distribution over $\{\mu_k, \Sigma_k\}_{k=1}^{K}$.

## F.1   M-step for $\nu_0$

Expanding the first objective yields,

$$\mathcal{L}(\nu_0) = \sum_{k=1}^{K} \mathbb{E}\left[\log\left\{\frac{|(\nu_0 - D - 1)\Sigma_0|^{\frac{\nu_0}{2}}}{2^{\frac{\nu_0 D}{2}}\Gamma_D(\frac{\nu_0}{2})}|\Sigma_k|^{-\frac{\nu_0+D+1}{2}}e^{-\text{Tr}(\frac{\nu_0-D-1}{2}\Sigma_0\Sigma_k^{-1})}\right\}\right]$$

$$= \sum_{k=1}^{K} \frac{\nu_0 D}{2}\log\left(\frac{\nu_0-D-1}{2}\right) + \frac{\nu_0}{2}\log|\Sigma_0| - \log\Gamma_D(\frac{\nu_0}{2}) - \frac{\nu_0+D+1}{2}\mathbb{E}[\log|\Sigma_k|] - \frac{\nu_0-D-1}{2}\text{Tr}(\Sigma_0\mathbb{E}[\Sigma_k^{-1}])$$

$$= \sum_{k=1}^{K} \frac{\nu_0 D}{2}\log\left(\frac{\nu_0-D-1}{2}\right) + \frac{\nu_0}{2}\left[\log|\Sigma_0| - \mathbb{E}[\log|\Sigma_k|] - \text{Tr}(\Sigma_0\mathbb{E}[\Sigma_k^{-1}])\right] - \log\Gamma_D(\frac{\nu_0}{2}).$$

We can maximize this objective using a generalized Newton's method (Minka, 2000). We need the first and second derivatives of the objective,

$$\mathcal{L}'(\nu_0) = \sum_{k=1}^{K} \frac{D}{2}\left[\log\left(\frac{\nu_0-D-1}{2}\right) + \frac{\nu_0}{\nu_0-D-1}\right] + \frac{1}{2}\left[\log|\Sigma_0| - \mathbb{E}[\log|\Sigma_k|] - \text{Tr}(\Sigma_0\mathbb{E}[\Sigma_k^{-1}])\right] - \frac{1}{2}\psi_D(\frac{\nu_0}{2})$$

$$\mathcal{L}''(\nu_0) = \sum_{k=1}^{K} \frac{D}{2}\left[\frac{1}{\nu_0-D-1} - \frac{D+1}{(\nu_0-D-1)^2}\right] - \frac{1}{4}\psi_D^{(2)}(\frac{\nu_0}{2}).$$

The idea is to lower bound the objective with a concave function of the form,

$$g(\nu_0) = k + a \log \nu_0 + b\nu_0$$

which has derivatives $g'(\nu_0) = \frac{a}{\nu_0} + b$ and $g''(\nu_0) = -\frac{a}{\nu_0^2}$. Matching derivatives implies,

$$a = -\nu_0^2 \mathcal{L}''(\nu_0)$$
$$b = \mathcal{L}'(\nu_0) - \frac{a}{\nu_0}$$
$$k = \mathcal{L}(\nu_0) - a \log \nu_0 - b\nu_0.$$

For $a > 0$ and $b < 0$, the maximizer of the lower bound is obtained at

$$\nu_0^\star = -\frac{a}{b} = \frac{\nu_0^2 \mathcal{L}''(\nu_0)}{\mathcal{L}'(\nu_0) + \nu_0 \mathcal{L}''(\nu_0)} \tag{20}$$

## F.2 M-step for $\kappa_0$

Expanding the second objective,

$$\mathcal{L}(\kappa_0) = \sum_{k=1}^{K} \frac{D}{2} \log \kappa_0 - \frac{\kappa_0}{2} \mathbb{E}\left[(\mu_k - \mu_0)^\top \Sigma_k^{-1} (\mu_k - \mu_0)\right] + c.$$

The maximum is obtained at,

$$\kappa_0^\star = \left(\frac{1}{KD} \sum_{k=1}^{K} \mathbb{E}\left[(\mu_k - \mu_0)^\top \Sigma_k^{-1} (\mu_k - \mu_0)\right]\right)^{-1}.$$

## F.3 Computing the posterior expectations

Under the conjugate prior, those posteriors are normal inverse Wishart distributions,

$$\mu_k, \Sigma_k \mid \{x_n : y_n = k\} \sim \mathrm{NIW}(\nu_k', \Sigma_k', \kappa_k', \mu_k')$$
$$\nu_k' = \nu_0 + N_k$$
$$\kappa_k' = \kappa_0 + N_k$$
$$\mu_k' = \frac{1}{\kappa_k'} \left(\kappa_0 \mu_0 + \sum_{n:y_n=k} x_n\right)$$
$$\Sigma_k' = (\nu_0 - D - 1)\Sigma_0 + \kappa_0 \mu_0 \mu_0^\top + \sum_{n:y_n=k} x_n x_n^\top - \kappa_k' \mu_k' \mu_k'^\top$$

To evaluate the objectives above, we need the following expected sufficient statistics of the normal inverse Wishart distribution,

$$\mathbb{E}[\Sigma_k^{-1}] = \nu_k' \Sigma_k'^{-1}$$
$$\mathbb{E}[\log|\Sigma_k|] = \log|\Sigma_k'| - \psi_D(\tfrac{\nu_k'}{2}) - D \log 2$$
$$\mathbb{E}[(\mu_k - \mu_0)^\top \Sigma_k^{-1} (\mu_k - \mu_0)] = \frac{1}{\kappa_k'} + (\mu_k' - \mu_0)^\top \mathbb{E}[\Sigma_k^{-1}](\mu_k' - \mu_0)$$

Since the tied covariance approach in RMDS already works quite well, we recommend initializing the EM iterations by setting $\nu_0 \approx \bar{N}_k$ and $\kappa_0 \approx 0$. That way, the covariances are strongly coupled across clusters and the means have an uninformative prior.

### F.4 Marginal Likelihood

This EM algorithm maximizes the marginal likelihood,

$$
\begin{aligned}
\log p(\{x_n, y_n\}_{n=1}^N) &= \sum_{k=1}^K \log p(\{x_n : y_n = k\}) \\
&= \sum_{k=1}^K \log \int p(\{x_n : y_n = k\} \mid \mu_k, \Sigma_k)\, p(\mu_k, \Sigma_k)\, \mathrm{d}\mu_k\, \mathrm{d}\Sigma_k \\
&= \sum_{k=1}^K \log \int \left[ \prod_{n:y_n=k} \mathrm{N}(x_n \mid \mu_k, \Sigma_k) \right] \mathrm{NIW}(\mu_k, \Sigma_k \mid \nu_0, \Sigma_0, \kappa_0, \mu_0)\, \mathrm{d}\mu_k\, \mathrm{d}\Sigma_k \\
&= \sum_{k=1}^K \log Z(\nu_k', \Sigma_k', \kappa_k', \mu_k') - \log Z(\nu_0, (\nu_0 - D - 1)\Sigma_0, \kappa_0, \mu_0) + c
\end{aligned}
$$

where

$$
\log Z(\nu, \Sigma, \kappa, \mu) = -\tfrac{D}{2} \log \kappa + \log \Gamma_D(\tfrac{\nu}{2}) + \tfrac{\nu D}{2} \log 2 - \tfrac{\nu}{2} \log |\Sigma|
$$

is the log normalizer of the normal inverse Wishart distribution, and $c$ is constant with respect to the hyperparameters being optimized (but it is data dependent).

## G  EM Algorithm for the Diagonal Covariance Model

Here we describe an expectation-maximization (EM) algorithm for estimating the hyperparameters of the hierarchical diagonal covariance model. Recall that under this model,

$$
x_{n,d} \mid y_n = k \sim \mathrm{N}(\mu_{k,d}, \sigma_{k,d}^2)
$$

where

$$
\begin{aligned}
\sigma_{k,d}^2 &\sim \chi^{-2}(\nu_{0,d}, \sigma_{0,d}^2) \\
\mu_{k,d} &\sim \mathrm{N}(\mu_{0,d}, \kappa_{0,d}^{-1}\sigma_{k,d}^2),
\end{aligned}
$$

for each dimension $d = 1, \ldots, D$ independently. Under the prior $\mathbb{E}[\sigma_{k,d}^2] = \frac{\nu_{0,d}}{\nu_{0,d}-2}\sigma_{0,d}^2$, which is approximately $\sigma_{0,d}^2$ for large degrees of freedom $\nu_{0,d}$.

First, we set the hyperparameters $\mu_{0,d}$ and $\sigma_{0,d}^2$ using empirical Bayes estimates,

$$
\mu_{0,d} = \hat{\mu}_{0,d} = \frac{1}{N} \sum_{n=1}^N x_{n,d} \tag{21}
$$

$$
\sigma_{0,d}^2 = \hat{\sigma}_d^2 = \frac{1}{N} \sum_{n=1}^N (x_{n,d} - \hat{\mu}_{y_n,d})(x_{n,d} - \hat{\mu}_{y_n,d})^\top \tag{22}
$$

where $\hat{\mu}_{y_n,d} = \frac{1}{N_k} \sum_{n:y_n=k} x_{n,d}$.

To set the remaining hyperparameters, $\kappa_{0,d}$ and $\nu_{0,d}$, we use EM. The expected log likelihood is separable over these two hyperparameters,

$$
\mathcal{L}(\nu_{0,d}, \kappa_{0,d}) = \mathcal{L}(\nu_{0,d}) + \mathcal{L}(\kappa_{0,d})
$$

$$
\mathcal{L}(\nu_{0,d}) = \mathbb{E}\left[ \sum_{k=1}^K \log \chi^{-2}(\sigma_{k,d}^2 \mid \nu_{0,d}, \sigma_{0,d}^2) \right]
$$

$$
\mathcal{L}(\kappa_{0,d}) = \mathbb{E}\left[ \sum_{k=1}^K \log \mathrm{N}(\mu_{k,d} \mid \mu_{0,d}, \kappa_{0,d}^{-1}\sigma_{k,d}^2) \right]
$$

where the expectations are taken with respect to the posterior distribution over $\{\mu_{k,d}, \sigma_{k,d}\}_{k=1}^K$.

### G.1  M-step for $\nu_{0,d}$

Expanding the first objective yields,

$$
\begin{aligned}
\mathcal{L}(\nu_{0,d}) &= \sum_{k=1}^{K} \mathbb{E}\left[\log\left\{\frac{\left(\frac{\nu_{0,d}}{2}\sigma_{0,d}^2\right)^{\frac{\nu_{0,d}}{2}}}{\Gamma(\frac{\nu_{0,d}}{2})}(\sigma_{k,d}^2)^{-\frac{\nu_{0,d}+2}{2}}e^{-\frac{\nu_{0,d}\sigma_{0,d}^2}{2\sigma_{k,d}^2}}\right\}\right] \\
&= \sum_{k=1}^{K} \frac{\nu_{0,d}}{2}\log\left(\frac{\nu_{0,d}}{2}\right) + \frac{\nu_{0,d}}{2}\log\sigma_{0,d}^2 - \log\Gamma(\tfrac{\nu_{0,d}}{2}) - \frac{\nu_{0,d}+2}{2}\mathbb{E}[\log\sigma_{k,d}^2] - \frac{\nu_{0,d}}{2}\sigma_{0,d}^2\mathbb{E}[\sigma_{k,d}^{-2}] \\
&= \sum_{k=1}^{K} \frac{\nu_{0,d}}{2}\log\left(\frac{\nu_{0,d}}{2}\right) + \frac{\nu_{0,d}}{2}\left[\log\sigma_{0,d}^2 - \mathbb{E}[\log\sigma_{k,d}^2] - \sigma_{0,d}^2\mathbb{E}[\sigma_{k,d}^{-2}])\right] - \log\Gamma(\tfrac{\nu_{0,d}}{2}) + c.
\end{aligned}
$$

We can maximize this objective using a generalized Newton's method (Minka, 2000). We need the first and second derivatives of the objective,

$$
\mathcal{L}'(\nu_{0,d}) = \sum_{k=1}^{K} \tfrac{1}{2}\left[\log\left(\tfrac{\nu_{0,d}}{2}\right)+1\right] + \tfrac{1}{2}\left[\log\sigma_{0,d}^2 - \mathbb{E}[\log\sigma_{k,d}^2] - \sigma_{0,d}^2\mathbb{E}[\sigma_{k,d}^{-2}])\right] - \tfrac{1}{2}\psi(\tfrac{\nu_{0,d}}{2})
$$

$$
\mathcal{L}''(\nu_{0,d}) = \sum_{k=1}^{K} \tfrac{1}{2\nu_{0,d}} - \tfrac{1}{4}\psi'(\tfrac{\nu_{0,d}}{2}).
$$

The idea is to lower bound the objective with a concave function of the form,

$$
g(\nu_{0,d}) = k + a\log\nu_{0,d} + b\nu_{0,d}
$$

which has derivatives $g'(\nu_{0,d}) = \frac{a}{\nu_{0,d}} + b$ and $g''(\nu_{0,d}) = -\frac{a}{\nu_{0,d}^2}$. Matching derivatives implies,

$$
\begin{aligned}
a &= -\nu_{0,d}^2\mathcal{L}''(\nu_{0,d}) \\
b &= \mathcal{L}'(\nu_{0,d}) - \tfrac{a}{\nu_{0,d}} \\
k &= \mathcal{L}(\nu_{0,d}) - a\log\nu_{0,d} - b\nu_{0,d}.
\end{aligned}
$$

For $a > 0$ and $b < 0$, the maximizer of the lower bound is obtained at

$$
\nu_{0,d}^{\star} = -\frac{a}{b} = \frac{\nu_{0,d}^2\mathcal{L}''(\nu_{0,d})}{\mathcal{L}'(\nu_{0,d}) + \nu_{0,d}\mathcal{L}''(\nu_{0,d})} \tag{23}
$$

### G.2  M-step for $\kappa_{0,d}$

Expanding the second objective,

$$
\mathcal{L}(\kappa_{0,d}) = \sum_{k=1}^{K} \tfrac{1}{2}\log\kappa_{0,d} - \frac{\kappa_{0,d}}{2}\mathbb{E}\left[\frac{(\mu_{k,d}-\mu_{0,d})^2}{\sigma_{k,d}^2}\right] + c.
$$

The maximum is obtained at,

$$
\kappa_{0,d}^{\star} = \left(\frac{1}{K}\sum_{k=1}^{K}\mathbb{E}\left[\frac{(\mu_{k,d}-\mu_{0,d})^2}{\sigma_{k,d}^2}\right]\right)^{-1}.
$$

### G.3 Computing the posterior expectations

Under the conjugate prior, those posteriors are normal inverse chi-squared distributions,

$$\mu_{k,d}, \sigma_{k,d}^2 \mid \{x_n : y_n = k\} \sim \text{NIX}(\nu'_{k,d}, \sigma'^2_{k,d}, \kappa'_{k,d}, \mu'_{k,d})$$

$$\nu'_{k,d} = \nu_{0,d} + N_k$$

$$\kappa'_{k,d} = \kappa_{0,d} + N_k$$

$$\mu'_{k,d} = \frac{1}{\kappa'_{k,d}} \left( \kappa_0 \mu_{0,d} + \sum_{n:y_n=k} x_n \right)$$

$$\sigma'^2_{k,d} = \frac{1}{\nu'_{k,d}} \left[ \nu_{0,d} \sigma_{0,d}^2 + \kappa_{0,d} \mu_{0,d}^2 + \sum_{n:y_n=k} x_{n,d}^2 - \kappa'_{k,d} \mu'^2_{k,d} \right]$$

To evaluate the objectives above, we need the following expected sufficient statistics of the normal inverse chi-squared distribution,

$$\mathbb{E}[\sigma_{k,d}^{-2}] = \sigma'^{-2}_{k,d}$$

$$\mathbb{E}[\log \sigma_{k,d}^2] = \log \frac{\nu'_{k,d} \sigma'^2_{k,d}}{2} - \psi(\frac{\nu'_{k,d}}{2})$$

$$\mathbb{E}\left[ \frac{(\mu_{k,d} - \mu_{0,d})^2}{\sigma_{k,d}^2} \right] = \frac{1}{\kappa'_{k,d}} + \frac{(\mu'_{k,d} - \mu_{0,d})^2}{\sigma'^2_{k,d}}$$

Since the tied covariance approach in RMDS already works quite well, we recommend initializing the EM iterations by setting $\nu_{0,d} \approx \bar{N}_k$ and $\kappa_{0,d} \approx 0$. That way, the covariances are strongly coupled across clusters and the means have an uninformative prior.

### G.4 Marginal Likelihood

This EM algorithm maximizes the marginal likelihood,

$$\log p(\{x_n, y_n\}_{n=1}^N) = \sum_{k=1}^K \log p(\{x_n : y_n = k\})$$

$$= \sum_{d=1}^D \sum_{k=1}^K \log \int p(\{x_{n,d} : y_n = k\} \mid \mu_{k,d}, \sigma_{k,d}^2) \, p(\mu_{k,d} \sigma_{k,d}^2) \, \mathrm{d}\mu_{k,d} \, \mathrm{d}\sigma_{k,d}^2$$

$$= \sum_{d=1}^D \sum_{k=1}^K \log \int \left[ \prod_{n:y_n=k} \text{N}(x_{n,d} \mid \mu_{k,d}, \sigma_{k,d}^2) \right] \text{NIX}(\mu_{k,d}, \sigma_{k,d}^2 \mid \nu_{0,d}, \sigma_{0,d}^2, \kappa_{0,d}, \mu_{0,d}) \, \mathrm{d}\mu_{k,d} \, \mathrm{d}\sigma_{k,d}^2$$

$$= \sum_{d=1}^D \left[ \sum_{k=1}^K \left( \log Z(\nu'_{k,d}, \sigma'^2_{k,d}, \kappa'_{k,d}, \mu'_{k,d}) - \log Z(\nu_{0,d}, \sigma_{0,d}^2, \kappa_{0,d}, \mu_{0,d}) \right) + c \right]$$

where

$$\log Z(\nu, \sigma^2, \kappa, \mu) = -\tfrac{1}{2} \log \kappa + \log \Gamma(\tfrac{\nu}{2}) - \tfrac{\nu}{2} \log \tfrac{\nu \sigma^2}{2}$$

is the log normalizer of the normal inverse chi-squared distribution, and $c = \log(2\pi)^{-N/2}$ is constant with respect to the hyperparameters being optimized.

### G.5 Predictive Distributions

Under this model, the predictive distribution of a data point given its cluster assignment is,

$$
\begin{aligned}
p(x \mid y = k, X_{\mathsf{tr}}, y_{\mathsf{tr}}) &= \prod_{d=1}^{D} \int p(x_d \mid \mu_{k,d}, \sigma_{k,d}^2)\, p(\mu_{k,d}, \sigma_{k,d}^2 \mid X_{\mathsf{tr}}, y_{\mathsf{tr}})\, \mathrm{d}\mu_{k,d}\, \mathrm{d}\sigma_{k,d}^2 \\
&= \prod_{d=1}^{D} \int \mathrm{N}(x_d \mid \mu_{k,d}, \sigma_{k,d}^2)\, \mathrm{NIX}(\mu_{k,d}, \sigma_{k,d}^2 \mid \nu'_{k,d}, \sigma'^2_{k,d}, \kappa'_{k,d}, \mu'_{k,d})\, \mathrm{d}\mu_{k,d}\, \mathrm{d}\sigma_{k,d}^2 \\
&= \prod_{d=1}^{D} \mathrm{St}(x_d \mid \nu'_{k,d}, \mu'_{k,d}, \tfrac{\kappa'_{k,d}+1}{\kappa'_{k,d}}\sigma'^2_{k,d}),
\end{aligned}
$$

where $\mathrm{St}(x \mid \nu, \mu, \sigma^2)$ denotes the univariate Student's t distribution with $\nu$ degrees of freedom, location $\mu$, and scale $\sigma$.

Under this model, the prior predictive distribution is,

$$
p(x \mid y = K+1, X_{\mathsf{tr}}, y_{\mathsf{tr}}) = \prod_{d=1}^{D} \mathrm{St}(x_d \mid \nu_{0,d}, \mu_{0,d}, \tfrac{\kappa_{0,d}+1}{\kappa_{0,d}}\sigma_{0,d}^2),
$$

which is approximately Gaussian, $\mathrm{N}(x_d \mid \mu_{0,d}, \sigma_{0,d}^2)$ when $\nu_{0,d}, \kappa_{0,d} \gg 1$.

## H  EM Algorithm for the Coupled Diagonal Covariance Model

This model introduces a scale factor $\gamma_k \in \mathbb{R}_+$ that is shared by all dimensions. The model is,

$$
\begin{aligned}
\gamma_k &\sim \chi^2(\alpha_0) \\
\sigma_{k,d} &\sim \chi^{-2}(\nu_{0,d}, \gamma_k \sigma_{0,d}^2) && \text{for } d = 1, \ldots D \\
\mu_{k,d} &\sim \mathrm{N}(\mu_{0,d}, \kappa_{0,d}^{-1}\sigma_{k,d}^2) && \text{for } d = 1, \ldots D
\end{aligned}
$$

Since $\mathbb{E}[\gamma_k] = 1$, under the prior $\mathbb{E}[\sigma_{k,d}^2] = \frac{\nu_{0,d}}{\nu_{0,d}-2}\sigma_{0,d}^2$, which is approximately $\sigma_{0,d}^2$ for large $\nu_{0,d}$.

The hyperparameters of the model are $\eta = (\alpha_0, \{\nu_{0,d}, \sigma_{0,d}^2, \kappa_{0,d}, \mu_{0,d}\})$. We set the hyperparameters $\mu_{0,d}$ and $\sigma_{0,d}^2$ using empirical Bayes estimates,

$$
\mu_{0,d} = \hat{\mu}_{0,d} = \frac{1}{N}\sum_{n=1}^{N} x_{n,d} \tag{24}
$$

$$
\sigma_{0,d}^2 = \hat{\sigma}_d^2 = \frac{1}{N}\sum_{n=1}^{N} (x_{n,d} - \hat{\mu}_{y_n,d})^2 \tag{25}
$$

where $\hat{\mu}_k = \frac{1}{N_k}\sum_{n:y_n=k} x_n$ and $N_k = \sum_n \mathbb{I}[y_n = k]$.

To set the remaining hyperparameters, we use EM.

### H.1  E-step

Note that the posterior distribution factors as,

$$
\begin{aligned}
p(\gamma_k, \{\mu_{k,d}, \sigma_{k,d}^2\}_{d=1}^D \mid X_k) &= p(\gamma_k \mid X_k)\, p(\{\mu_{k,d}, \sigma_{k,d}^2\}_{d=1}^D \mid \gamma_k, X_k) \\
&= p(\gamma_k \mid X_k) \prod_{d=1}^{D} p(\mu_{k,d}, \sigma_{k,d}^2 \mid \gamma_k, X_k).
\end{aligned}
$$

The posterior distribution over $\gamma_k$ doesn't have a simple closed form, but since it's only one-dimensional, we can approximate it on a dense grid of points, $\{\gamma^{(p)}\}_{p=1}^P$. Conditioned on $\gamma_k = \gamma^{(p)}$, the distribution of $\mu_{k,d}$ and $\sigma_{k,d}^2$ is a normal inverse chi-squared. For each point,

$$p(\mu_{k,d}, \sigma_{k,d}^2 \mid \gamma_k = \gamma^{(p)}, X_k) = \text{NIX}(\mu_{k,d}, \sigma_{k,d}^2 \mid \nu_{k,d}', \sigma_{k,p,d}'^2, \kappa_{k,d}', \mu_{k,d}')$$

$$\nu_{k,d}' = \nu_{0,d} + N_k$$

$$\kappa_{k,d}' = \kappa_{0,d} + N_k$$

$$\mu_{k,d}' = \frac{1}{\kappa_{k,d}'} \left( \kappa_0 \mu_{0,d} + \sum_{n:y_n=k} x_n \right)$$

$$\sigma_{k,p,d}'^2 = \frac{1}{\nu_{k,d}'} \left[ \nu_{0,d} \gamma^{(p)} \sigma_{0,d}^2 + \kappa_{0,d} \mu_{0,d}^2 + \sum_{n:y_n=k} x_{n,d}^2 - \kappa_{k,d}' \mu_{k,d}'^2 \right]$$

Note that this is practically the same as above, but with $\gamma_k$ scaling the prior for $\sigma_{k,d}^2$. For any value of $\gamma_k$, the posterior probability is,

$$p(\gamma_k = \gamma^{(p)} \mid X_k) \propto p(\gamma_k = \gamma^{(p)}) \, p(X_k \mid \gamma_k = \gamma^{(p)})$$

$$= p(\gamma_k = \gamma^{(p)}) \prod_{d=1}^D p(X_{k,d} \mid \gamma_k = \gamma^{(p)})$$

$$= p(\gamma_k = \gamma^{(p)}) \prod_{d=1}^D \frac{Z(\nu_{k,d}', \sigma_{k,p,d}'^2, \kappa_{k,d}', \mu_{k,d}')}{Z(\nu_{0,d}, \gamma^{(p)} \sigma_{0,d}^2, \kappa_{0,d}, \mu_{0,d})}$$

$$\triangleq \tilde{w}_{k,p}'.$$

where we reused the marginal likelihood calculation from the hierarchical diagaonal DPMM above. Finally, denote the normalized posterior probabilities as,

$$w_{k,p}' = \frac{\tilde{w}_{k,p}'}{\sum_r \tilde{w}_{k,r}'}.$$

## H.2 M-Step

To set the hyperparameters, $\kappa_{0,d}$, $\nu_{0,d}$, and $\alpha_0$, we use EM. The expected log likelihood is separable over these two hyperparameters,

$$\mathcal{L}(\nu_{0,d}, \kappa_{0,d}, \alpha_0) = \mathcal{L}(\nu_{0,d}) + \mathcal{L}(\kappa_{0,d}) + \mathcal{L}(\alpha_0)$$

$$\mathcal{L}(\nu_{0,d}) = \mathbb{E} \left[ \sum_{k=1}^K \log \chi^{-2}(\sigma_{k,d}^2 \mid \nu_{0,d}, \gamma_k \sigma_{0,d}^2) \right]$$

$$\mathcal{L}(\kappa_{0,d}) = \mathbb{E} \left[ \sum_{k=1}^K \log \text{N}(\mu_{k,d} \mid \mu_{0,d}, \kappa_{0,d}^{-1} \sigma_{k,d}^2) \right]$$

$$\mathcal{L}(\alpha_0) = \mathbb{E} \left[ \log \text{Ga}(\gamma_k \mid \alpha_0, \alpha_0) \right]$$

where the expectations are taken with respect to the posterior distribution over $\{\gamma_k, \{\mu_{k,d}, \sigma_{k,d}\}_{d=1}^D\}_{k=1}^K$.

### H.3    M-step for $\nu_{0,d}$

Expanding the first objective yields,

$$\mathcal{L}(\nu_{0,d}) = \sum_{k=1}^{K} \mathbb{E}\left[\log\left\{\frac{(\frac{\nu_{0,d}}{2}\gamma_k\sigma_{0,d}^2)^{\frac{\nu_{0,d}}{2}}}{\Gamma(\frac{\nu_{0,d}}{2})}(\sigma_{k,d}^2)^{-\frac{\nu_{0,d}+2}{2}}e^{-\frac{\nu_{0,d}\gamma_k\sigma_{0,d}^2}{2\sigma_{k,d}^2}}\right\}\right]$$

$$= \sum_{k=1}^{K}\frac{\nu_{0,d}}{2}\log\left(\frac{\nu_{0,d}}{2}\right) + \frac{\nu_{0,d}}{2}\mathbb{E}[\log\gamma_k] + \frac{\nu_{0,d}}{2}\log\sigma_{0,d}^2 - \log\Gamma(\frac{\nu_{0,d}}{2}) - \frac{\nu_{0,d}+2}{2}\mathbb{E}[\log\sigma_{k,d}^2] - \frac{\nu_{0,d}}{2}\sigma_{0,d}^2\mathbb{E}[\gamma_k\sigma_{k,d}^{-2}]$$

$$= \sum_{k=1}^{K}\frac{\nu_{0,d}}{2}\log\left(\frac{\nu_{0,d}}{2}\right) + \frac{\nu_{0,d}}{2}\left[\mathbb{E}[\log\gamma_k] + \log\sigma_{0,d}^2 - \mathbb{E}[\log\sigma_{k,d}^2] - \sigma_{0,d}^2\mathbb{E}[\gamma_k\sigma_{k,d}^{-2}])\right] - \log\Gamma(\frac{\nu_{0,d}}{2}) + c.$$

We can maximize this objective using a generalized Newton's method (Minka, 2000). We need the first and second derivatives of the objective,

$$\mathcal{L}'(\nu_{0,d}) = \sum_{k=1}^{K}\frac{1}{2}\left[\log\left(\frac{\nu_{0,d}}{2}\right) + 1\right] + \frac{1}{2}\left[\mathbb{E}[\log\gamma_k] + \log\sigma_{0,d}^2 - \mathbb{E}[\log\sigma_{k,d}^2] - \sigma_{0,d}^2\mathbb{E}[\gamma_k\sigma_{k,d}^{-2}])\right] - \frac{1}{2}\psi(\frac{\nu_{0,d}}{2})$$

$$\mathcal{L}''(\nu_{0,d}) = \sum_{k=1}^{K}\frac{1}{2\nu_{0,d}} - \frac{1}{4}\psi'(\frac{\nu_{0,d}}{2}).$$

The idea is to lower bound the objective with a concave function of the form,

$$g(\nu_{0,d}) = k + a\log\nu_{0,d} + b\nu_{0,d}$$

which has derivatives $g'(\nu_{0,d}) = \frac{a}{\nu_{0,d}} + b$ and $g''(\nu_{0,d}) = -\frac{a}{\nu_{0,d}^2}$. Matching derivatives implies,

$$a = -\nu_{0,d}^2\mathcal{L}''(\nu_{0,d})$$
$$b = \mathcal{L}'(\nu_{0,d}) - \frac{a}{\nu_{0,d}}$$
$$k = \mathcal{L}(\nu_{0,d}) - a\log\nu_{0,d} - b\nu_{0,d}.$$

For $a > 0$ and $b < 0$, the maximizer of the lower bound is obtained at

$$\nu_{0,d}^{\star} = -\frac{a}{b} = \frac{\nu_{0,d}^2\mathcal{L}''(\nu_{0,d})}{\mathcal{L}'(\nu_{0,d}) + \nu_{0,d}\mathcal{L}''(\nu_{0,d})} \tag{26}$$

### H.4    M-step for $\kappa_{0,d}$

Expanding the second objective,

$$\mathcal{L}(\kappa_{0,d}) = \sum_{k=1}^{K}\frac{1}{2}\log\kappa_{0,d} - \frac{\kappa_{0,d}}{2}\mathbb{E}\left[\frac{(\mu_{k,d}-\mu_{0,d})^2}{\sigma_{k,d}^2}\right] + c.$$

The maximum is obtained at,

$$\kappa_{0,d}^{\star} = \left(\frac{1}{K}\sum_{k=1}^{K}\mathbb{E}\left[\frac{(\mu_{k,d}-\mu_{0,d})^2}{\sigma_{k,d}^2}\right]\right)^{-1}.$$

### H.5    M-step for $\alpha_0$

Expanding the final objective,

$$\mathcal{L}(\alpha_0) = \sum_{k=1}^{K}\alpha_0\log\alpha_0 - \log\Gamma(\alpha_0) + \alpha_0\mathbb{E}[\log\gamma_k] - \alpha_0\mathbb{E}[\gamma_k] + c.$$

Its derivatives are,

$$\mathcal{L}'(\alpha_0) = K \log \alpha_0 + K - K\psi(\alpha_0) + \sum_{k=1}^{K} \mathbb{E}\left[\log \gamma_k\right] - \mathbb{E}[\gamma_k]$$

$$\mathcal{L}''(\alpha_0) = \frac{K}{\alpha_0} - K\psi'(\alpha_0)$$

We can optimize $\alpha_0$ using the generalized Newton's method described in the M-step for $\nu_0$.

### H.6 Computing the posterior expectations

To evaluate the objectives above, we need the following expected sufficient statistics of the normal inverse chi-squared distribution,

$$\mathbb{E}[\gamma_k] = \sum_p w'_{k,p}\gamma^{(p)}$$

$$\mathbb{E}[\log \gamma_k] = \sum_p w'_{k,p} \log \gamma^{(p)}$$

$$\mathbb{E}[\gamma_k \sigma_{k,d}^{-2}] = \mathbb{E}_{\gamma_k}[\mathbb{E}_{\sigma_{k,d}^2|\gamma_k}[\gamma_k \sigma_{k,d}^{-2}]] = \sum_p w'_{k,p}\gamma^{(p)}\sigma_{k,p,d}'^{-2}$$

$$\mathbb{E}[\log \sigma_{k,d}^2] = \mathbb{E}_{\gamma_k}[\mathbb{E}_{\sigma_{k,d}^2|\gamma_k}[\log \sigma_{k,d}^2]] = \sum_p w'_{k,p}[\log \tfrac{\nu'_{k,d}\sigma'^2_{k,p,d}}{2} - \psi(\tfrac{\nu'_{k,d}}{2})]$$

$$\mathbb{E}\left[\frac{(\mu_{k,d} - \mu_{0,d})^2}{\sigma_{k,d}^2}\right] = \mathbb{E}_{\gamma_k}\left[\mathbb{E}_{\mu_{k,d},\sigma_{k,d}^2|\gamma_k}\left[\frac{(\mu_{k,d} - \mu_{0,d})^2}{\sigma_{k,d}^2}\right]\right] = \sum_p w'_{k,p}\left[\frac{1}{\kappa'_{k,d}} + \frac{(\mu'_{k,d} - \mu_{0,d})^2}{\sigma'^2_{k,p,d}}\right]$$

where $\gamma^{(p)}$ are the centers of discretized posterior on $\gamma_k$ and $w_{k,p}$ are the corresponding weights.

Since the tied covariance approach in RMDS already works quite well, we recommend initializing the EM iterations by setting $\nu_{0,d} \approx \bar{N}_k$ and $\kappa_{0,d} \approx 0$. That way, the covariances are strongly coupled across clusters and the means have an uninformative prior.

### H.7 Computing the predictive distributions

The prior predictive is,

$$p(x^\star; \eta_0) =$$
$$\int \left[\prod_{d=1}^{D} \int \mathrm{N}(x_d^\star \mid \mu_d, \sigma_d^2)\mathrm{NIX}(\mu_d, \sigma_d^2 \mid \nu_{0,d}, \kappa_{0,d}, \mu_{0,d}, \gamma\sigma_{0,d}^2, \{x_n, y_n\}_{n=1}^{N}) \, d\mu_d \, d\sigma_d^2\right] \mathrm{Ga}(\gamma \mid \alpha_0, \alpha_0) \, d\gamma.$$

where $\eta_0 = \alpha_0, \{\mu_{0,d}, \sigma_{0,d}^2, \kappa_{0,d}, \nu_{0,d}\}_{d=1}^{D}$ are the model hyperparamters.

The $\gamma$ integral can be estimated by numerical integration over a dense grid of points,

$$p(x^\star; \eta_0) \approx \sum_{p=1}^{P} w_{0,p}\left[\prod_{d=1}^{D} \int \mathrm{N}(x_d^\star \mid \mu_d, \sigma_d^2)\mathrm{NIX}(\mu_d, \sigma_d^2 \mid \nu_{0,d}, \kappa_{0,d}, \mu_{0,d}, \gamma^{(p)}\sigma_{0,d}^2, \{x_n, y_n\}_{n=1}^{N}) \, d\mu_d \, d\sigma_d^2\right]$$

$$= \sum_{p=1}^{P} w_{0,p}\left[\prod_{d=1}^{D} \mathrm{St}(x_d^\star \mid \nu_{0,d}, \mu_{0,d}, \tfrac{\kappa_{0,d}+1}{\kappa_{0,d}}\gamma^{(p)}\sigma_{0,d}^2)\right]$$

where,

$$w_{0,p} = \frac{\mathrm{Ga}(\gamma^{(p)} \mid \alpha_0, \alpha_0)\Delta\gamma^{(p)}}{\sum_r \mathrm{Ga}(\gamma^{(r)} \mid \alpha_0, \alpha_0)\Delta\gamma^{(r)}}.$$

We renormalize the weights to ensure that the numerical integration satisfies that $\mathbb{E}_\gamma[1] = 1$.

In practice, we evaluate the prior *log* predictive probability using a log-sum-exp,

$$\log p(x^\star; \eta_0) \approx \text{logsumexp}_p \left[ \log w_{0,p} + \sum_{d=1}^{D} \log \text{St}(x_d^\star \mid \nu_{0,d}, \mu_{0,d}, \tfrac{\kappa_{0,d}+1}{\kappa_{0,d}} \gamma^{(p)} \sigma_{0,d}^2) \right]$$

By the same logic, the posterior log predictive is,

$$\log p(x^\star \mid y^\star = k, \{x_n, y_n\}_{n=1}^{N}; \eta_0) \approx \text{logsumexp}_p \left[ \log w_{k,p}' + \sum_{d=1}^{D} \log \text{St}(x_d^\star \mid \nu_{k,d}', \mu_{k,d}', \tfrac{\kappa_{k,d}'+1}{\kappa_{k,d}'} \gamma^{(p)} \sigma_{k,p,d}'^2) \right]$$

### H.8 Marginal Likelihood

This EM algorithm maximizes the marginal likelihood,

$$
\begin{aligned}
\log p(\{x_n, y_n\}_{n=1}^{N}) &= \sum_{k=1}^{K} \log p(\{x_n : y_n = k\}) \\
&= \sum_{k=1}^{K} \log \int p(\{x_n : y_n = k\} \mid \{\mu_{k,d}, \sigma_{k,d}^2\}) \, p(\{\mu_{k,d} \sigma_{k,d}^2\} \mid \gamma_k) \, p(\gamma_k) \, \mathrm{d}\mu_{k,d} \, \mathrm{d}\sigma_{k,d}^2 \, \mathrm{d}\gamma_k \\
&= \sum_{k} \log \int \left[ \prod_{d=1}^{D} \int \left[ \prod_{n:y_n=k} p(x_{n,d} \mid \mu_{k,d}, \sigma_{k,d}^2) p(\mu_{k,d}, \sigma_{k,d}^2 \mid \gamma_k) \, \mathrm{d}\mu_{k,d} \, \mathrm{d}\sigma_{k,d}^2 \right] p(\gamma_k) \, \mathrm{d}\gamma_k \right] \\
&= \sum_{k} \log \int \prod_{d=1}^{D} \left[ \frac{Z(\nu_{k,d}', \sigma_{k,p,d}'^2, \kappa_{k,d}', \mu_{k,d}')}{Z(\nu_{0,d}, \gamma_k \sigma_{0,d}^2, \kappa_{0,d}, \mu_{0,d})} \right] p(\gamma_k) \, \mathrm{d}\gamma_k + c \\
&\approx \sum_{k} \log \sum_{p} w_{0,p} \prod_{d=1}^{D} \frac{Z(\nu_{k,d}', \sigma_{k,p,d}'^2, \kappa_{k,d}', \mu_{k,d}')}{Z(\nu_{0,d}, \gamma^{(p)} \sigma_{0,d}^2, \kappa_{0,d}, \mu_{0,d})} + c \\
&= \sum_{k} \text{logsumexp}_p \left[ \ell_{k,p} \right] + c
\end{aligned}
$$

where

$$\ell_{k,p} \triangleq \log w_{0,p} + \sum_{d=1}^{D} \left( \log Z(\nu_{k,d}', \sigma_{k,p,d}'^2, \kappa_{k,d}', \mu_{k,d}') - \log Z(\nu_{0,d}, \gamma^{(p)} \sigma_{0,d}^2, \kappa_{0,d}, \mu_{0,d}) \right)$$

and $c = \frac{ND}{2} \log 2\pi$.

# I  Score Correlation between RMDS and the Tied Covariance Model

We observe that the DPMM model with shared covariance and the RMDS Ren et al. (2021) are highly correlated, as illustrated in Figure 6. Here, we plot the RMDS vs the Tied Covariance Gaussian DPMM for all the real datasets (differentiated by color) in the Imagenet-1K OpenOOD task and note the tight agreement between the two. This empirical result supports the theoretical relationship derived in Proposition 4.2.

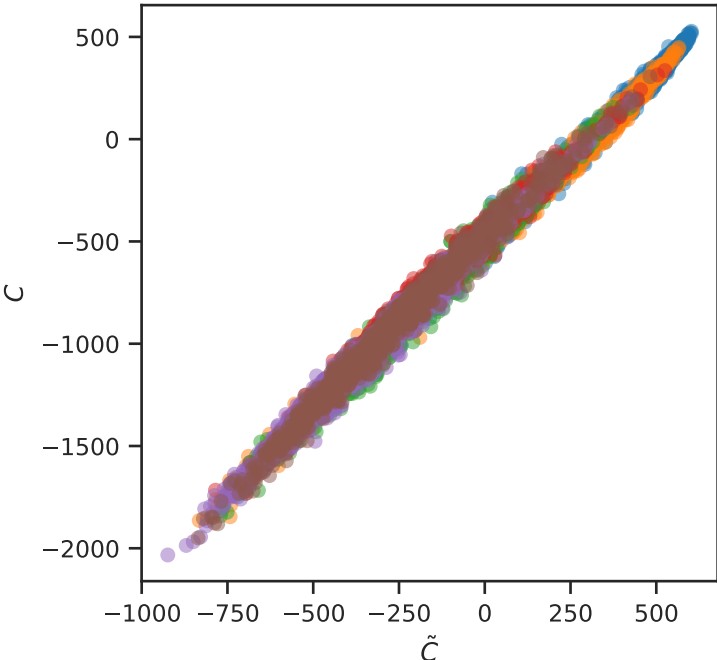

Figure 6: Tied DPMM OOD score, $\tilde{C}$, correlation to RMDS Ren et al. (2021) score, $C$, on the Imagenet-1K dataset. The colors represent different ID or OOD datasets.

## J Imagenet-1K Experiment

Table 2: OpenOOD performance across different preprocessing methods for expectation maximization trained hierarchical DPMM models. The preprocessing methods are the raw ViT features (ViT), marginal covariance whitening followed by a rotation into the average class-covariance eigenspace (W&R), and PCA. Baselines: Mahalanobis distance score to the closely related Mahalanobis distance score methods, MDS (Lee et al., 2018) and RMDS (Ren et al., 2021). We also compare to maximum softmax probability (MSP) (Hendrycks & Gimpel, 2017) and temperature scaled MSP with $T = 1000$ (Temp. Scale) (Guo et al., 2017) which is the ODIN (Liang et al., 2018) method without input preprocessing. A single linear layer was trained with gradient descent and supervised cross-entropy loss for the MSP and ODIN methods. Accuracy values marked with a asterisk* are the ViT classification layer's performance not the method itself. Additional benchmark performance metrics were obtained from the OpenOODv1.5 leaderboard (Zhang et al., 2024) including ASH (Djurisic et al., 2023) and ReAct (Sun et al., 2021).

| Model | Pre | Accuracy | Near | | | Far | | | |
| | | | SSB Hard | NINCO | Avg. | iNaturalist | OpenImage-O | Textures | Avg. |
| --- | --- | --- | --- | --- | --- | --- | --- | --- | --- |
| MSP | ViT | **80.94** | 73.80 | 82.72 | 78.26 | 92.08 | 88.67 | 88.39 | 89.71 |
| | W&R | 80.90 | 71.74 | 79.87 | 75.80 | 88.65 | 85.62 | 84.64 | 86.30 |
| | PCA | 80.91 | 73.67 | 82.94 | 78.31 | 92.08 | 88.60 | 88.34 | 89.67 |
| Temp. | ViT | **80.94** | **75.60** | 84.36 | 79.98 | 94.22 | 90.82 | **90.82** | 91.95 |
| MSP | W&R | 80.90 | 73.29 | 81.28 | 77.29 | 91.24 | 87.82 | 86.81 | 88.62 |
| T=1000 | PCA | 80.91 | 75.24 | 84.77 | 80.00 | 94.25 | 90.78 | 90.75 | 91.93 |
| MDS | ViT | 80.22 | 71.45 | 86.44 | 78.94 | 95.96 | 92.33 | 89.37 | 92.55 |
| | W&R | 80.41 | 71.45 | 86.48 | 78.97 | 96.00 | 92.34 | 89.38 | 92.57 |
| | PCA | 80.41 | 71.45 | 86.48 | 78.97 | 96.00 | 92.34 | 89.38 | 92.57 |
| RMDS | ViT | 80.22 | 72.78 | 87.18 | 79.98 | 96.00 | 92.23 | 89.28 | 92.50 |
| | W&R | 80.41 | 72.79 | 87.28 | 80.03 | **96.09** | 92.29 | 89.38 | 92.59 |
| | PCA | 80.41 | 72.79 | 87.28 | 80.03 | **96.09** | 92.29 | 89.38 | 92.59 |
| ASH | ViT | 81.14* | — | — | 53.21 | — | — | — | 51.56 |
| ReAct | ViT | 81.14* | | | 69.26 | | | | 85.69 |
| Hierarchical Gaussian DPMMs | | | | | | | | | |
| Tied | ViT | 80.40 | 71.79 | 86.75 | 79.27 | 95.99 | **92.40** | 89.71 | **92.70** |
| | W&R | 80.41 | 71.80 | 86.76 | 79.28 | 96.00 | **92.40** | 89.72 | **92.70** |
| | PCA | 80.40 | 71.79 | 86.75 | 79.27 | 96.00 | **92.40** | 89.70 | **92.70** |
| Full | ViT | 76.82 | 62.64 | 78.32 | 70.48 | 85.76 | 84.95 | 88.03 | 86.24 |
| | W&R | 76.78 | 62.84 | 78.48 | 70.66 | 85.88 | 85.03 | 88.02 | 86.31 |
| | PCA | 76.82 | 62.64 | 78.33 | 70.49 | 85.76 | 84.95 | 88.03 | 86.25 |
| Diag. | ViT | 75.96 | 72.38 | 85.96 | 79.17 | 94.14 | 90.18 | 87.20 | 90.51 |
| | W&R | 76.54 | 73.89 | 87.32 | 80.60 | 95.36 | 90.78 | 86.42 | 90.85 |
| | PCA | 75.76 | 71.99 | 85.52 | 78.75 | 93.91 | 90.18 | 87.39 | 90.49 |
| Coupled | ViT | 75.93 | 72.80 | 86.15 | 79.48 | 94.08 | 90.20 | 87.19 | 90.49 |
| Diag. | W&R | 76.52 | 74.47 | **87.48** | **80.98** | 95.51 | 90.63 | 86.02 | 90.72 |
| | PCA | 75.76 | 72.40 | 85.97 | 79.19 | 95.02 | 90.92 | 88.09 | 91.34 |

Table 3: Performance of Hierarchical Gaussian DPMM and baseline methods on the OpenOOD benchmark datasets Yang et al. (2022); Zhang et al. (2024), including both Near (SSB Hard (Vaze et al., 2022b) and NINCO (Bitterwolf et al., 2023)) and Far (iNaturalist (Van Horn et al., 2018), OpenImage-O (Wang et al., 2022), and Textures (Kylberg, 2011)) OOD datasets. The first column reports the accuracy of the classifiers on predicting the label $y \in [K]$ for in-distribution test data. Other columns report AUROC scores for OOD detection on OpenOOD benchmark datasets.

| Method | Accuracy | Near | | | Far | | | |
| | | SSB Hard | NINCO | Avg. | iNaturalist | OpenImage O | Textures | Avg. |
|---|---|---|---|---|---|---|---|---|
| MSP | **80.90** | 71.74 | 79.87 | 75.80 | 88.65 | 85.62 | 84.64 | 86.30 |
| Temp. Scale | **80.90** | 73.29 | 81.28 | 77.29 | 91.24 | 87.82 | 86.81 | 88.62 |
| MDS | 80.41 | 71.45 | 86.48 | 78.97 | 96.00 | 92.34 | 89.38 | 92.57 |
| RMDS | 80.41 | 72.79 | 87.28 | 80.03 | **96.09** | 92.29 | 89.38 | 92.59 |
| Hierarchical Gaussian DPMMs | | | | | | | | |
| Tied | 80.41 | 71.80 | 86.76 | 79.28 | 96.00 | **92.40** | **89.72** | **92.70** |
| Full | 76.78 | 62.84 | 78.48 | 70.66 | 85.88 | 85.03 | 88.02 | 86.31 |
| Diagonal | 76.54 | 73.89 | 87.32 | 80.60 | 95.36 | 90.78 | 86.42 | 90.85 |
| Coupled Diag. | 76.52 | **74.47** | **87.48** | **80.98** | 95.51 | 90.63 | 86.02 | 90.72 |

# K  CIFAR-10 Experiment

Table 4: OpenOOD CIFAR 10 performance across different preprocessing methods for expectation maximization trained hierarchical DPMM models. The preprocessing methods are the raw ResNet18 features and marginal covariance whitening followed by a rotation into the average class-covariance eigenspace (W&R). Baselines: Mahalanobis distance score to the closely related Mahalanobis distance score methods, MDS (Lee et al., 2018) and RMDS (Ren et al., 2021). We also compare to maximum softmax probability (MSP) (Hendrycks & Gimpel, 2017) and temperature scaled MSP with $T = 1000$ (Temp. Scale) (Guo et al., 2017) which is the ODIN (Liang et al., 2018) method without input preprocessing. A single linear layer was trained with gradient descent and supervised cross-entropy loss for the MSP and ODIN methods. Standard deviation across 3 ResNet18 training runs with different seeds is provided in parentheses and smaller font. Additional benchmark performance metrics were obtained from the OpenOODv1.5 leaderboard (Zhang et al., 2024) including ASH (Djurisic et al., 2023), ReAct (Sun et al., 2021), SCALE (Xu et al., 2024), and ReweightOOD (Regmi et al., 2024). †: Alternative training OOD method, *: Accuracy value inherited from ResNet18's classifier.

| Model | Pre | Accuracy | Near | | | Far | | | | |
| | | | CIFAR 100 | Tiny Imagenet | Avg. | MNIST | Places365 | SVHN | Textures | Avg. |
|---|---|---|---|---|---|---|---|---|---|---|
| MSP | ResNet18 | 95.01(0.29) | 87.24(0.44) | 88.92(0.34) | 88.08(0.38) | 92.68(1.32) | 89.42(0.71) | 91.46(0.66) | 89.76(0.95) | 90.83(0.38) |
| | W&R | 94.93(0.27) | 87.38(0.55) | 89.33(0.36) | 88.36(0.45) | 94.21(1.04) | 88.49(1.23) | 93.49(1.28) | 91.00(0.51) | 91.80(0.99) |
| Temp. MSP | ResNet18 | 95.01(0.29) | 86.51(0.72) | 88.78(0.57) | 87.65(0.64) | 94.07(2.12) | 89.57(0.87) | 91.88(1.27) | 89.14(1.19) | 91.16(0.79) |
| | W&R | 94.93(0.27) | 87.42(0.73) | 89.43(0.48) | 88.43(0.60) | 94.18(1.34) | 88.59(1.32) | **93.51**(1.95) | 91.08(0.80) | 91.84(1.32) |
| MDS | ResNet18 | 95.01(0.29) | 83.59(2.27) | 84.97(2.53) | 84.28(2.40) | 90.10(2.41) | 84.90(2.54) | 91.17(0.48) | 92.69(1.05) | 89.72(1.35) |
| | W&R | **95.04**(0.28) | 84.63(2.17) | 86.19(2.46) | 85.41(2.31) | 91.45(1.43) | 86.97(2.16) | 90.19(0.59) | 92.00(1.27) | 90.15(1.16) |
| RMDS | ResNet18 | 95.01(0.29) | 88.84(0.35) | 90.83(0.28) | 89.83(0.28) | 93.23(0.79) | 91.51(0.11) | 91.96(0.26) | 92.23(0.23) | 92.23(0.21) |
| | W&R | **95.04**(0.28) | 88.83(0.33) | 90.83(0.28) | 89.83(0.28) | 93.67(0.90) | 91.57(0.10) | 92.25(0.53) | 92.20(0.26) | 92.42(0.30) |
| ASH | ResNet18 | 95.06* | — | — | 75.27 | — | — | — | — | 78.49 |
| ReAct | ResNet18 | 95.06* | — | — | 87.11 | — | — | — | — | 90.42 |
| SCALE | ResNet18 | 95.06* | — | — | 82.55 | — | — | — | — | 86.39 |
| ReweightOOD† | ResNet18 | — | — | — | **91.86** | — | — | — | — | **98.05** |
| | | | Hierarchical Gaussian DPMMs | | | | | | | |
| Tied | ResNet18 | 95.02(0.28) | 88.53(0.38) | 90.40(0.58) | 89.47(0.48) | 93.91(0.26) | 90.83(0.51) | 93.51(0.39) | 94.52(0.14) | 93.19(0.09) |
| | W&R | **95.04**(0.28) | 88.83(0.33) | 90.83(0.28) | 89.83(0.28) | 93.67(0.90) | 91.57(0.10) | 92.25(0.53) | 92.20(0.26) | 92.42(0.30) |
| Full | ResNet18 | 95.00(0.25) | 89.36(0.10) | 91.46(0.07) | 90.41(0.05) | **94.71**(0.76) | 91.06(0.32) | 93.36(0.25) | 92.39(0.27) | 92.88(0.30) |
| | W&R | 94.95(0.26) | **89.69**(0.13) | **91.57**(0.32) | **90.63**(0.22) | 94.32(0.47) | **91.95**(0.20) | 93.37(0.43) | **94.35**(0.13) | **93.50**(0.20) |
| Diag. | ResNet18 | 94.84(0.26) | 89.07(0.16) | 90.93(0.26) | 90.00(0.21) | 92.63(0.42) | 91.34(0.49) | 91.13(0.48) | 92.11(0.49) | 91.80(0.39) |
| | W&R | 94.76(0.32) | 88.01(0.26) | 90.27(0.33) | 89.14(0.29) | 91.50(0.75) | 91.70(0.40) | 88.07(0.81) | 92.16(0.06) | 90.86(0.34) |
| Coupled Diag. | ResNet18 | 94.84(0.26) | 89.13(0.18) | 90.98(0.27) | 90.06(0.22) | 92.82(0.50) | 91.47(0.40) | 91.31(0.57) | 92.16(0.56) | 91.94(0.42) |
| | W&R | 94.76(0.32) | 87.17(0.19) | 89.43(0.26) | 88.30(0.23) | 91.11(0.07) | 90.84(0.35) | 88.07(0.49) | 92.78(0.17) | 90.70(0.20) |

Table 5: Performance of Hierarchical Gaussian DPMM and baseline methods on the OpenOOD CIFAR-10 benchmark.

| Model | Accuracy | Near | | | Far | | | | |
| | | CIFAR 100 | Tiny Imagenet | Avg. | MNIST | Places365 | SVHN | Textures | Avg. |
|---|---|---|---|---|---|---|---|---|---|
| MSP | 94.93 | 87.38 | 89.33 | 88.36 | 94.21 | 88.49 | 93.49 | 91.00 | 91.80 |
| Temp. MSP | 94.93 | 87.42 | 89.43 | 88.43 | 94.18 | 88.59 | **93.51** | 91.08 | 91.84 |
| MDS | **95.04** | 84.63 | 86.19 | 85.41 | 91.45 | 86.97 | 90.19 | 92.00 | 90.15 |
| RMDS | **95.04** | 88.83 | 90.83 | 89.83 | 93.67 | 91.57 | 92.25 | 92.20 | 92.42 |
| | | Hierarchical Gaussian DPMMs | | | | | | | |
| Tied | **95.04** | 88.83 | 90.83 | 89.83 | 93.67 | 91.57 | 92.25 | 92.20 | 92.42 |
| Full | 94.95 | **89.69** | **91.57** | **90.63** | 94.32 | **91.95** | 93.37 | **94.35** | **93.50** |
| Diag. | 94.76 | 88.01 | 90.27 | 89.14 | 91.50 | 91.70 | 88.07 | 92.16 | 90.86 |
| Coupled Diag. | 94.76 | 87.17 | 89.43 | 88.30 | 91.11 | 90.84 | 88.07 | 92.78 | 90.70 |

# L   CIFAR-100 Experiment

Table 6: OpenOOD CIFAR 100 performance across different preprocessing methods for expectation maximization trained hierarchical DPMM models. The preprocessing methods are the raw ResNet18 features and marginal covariance whitening followed by a rotation into the average class-covariance eigenspace (W&R). Baselines: Mahalanobis distance score to the closely related Mahalanobis distance score methods, MDS (Lee et al., 2018) and RMDS (Ren et al., 2021). We also compare to maximum softmax probability (MSP) (Hendrycks & Gimpel, 2017) and temperature scaled MSP with $T = 1000$ (Temp. Scale) (Guo et al., 2017) which is the ODIN (Liang et al., 2018) method without input preprocessing. A single linear layer was trained with gradient descent and supervised cross-entropy loss for the MSP and ODIN methods. Additional benchmark performance metrics were obtained from the OpenOODv1.5 leaderboard (Zhang et al., 2024) including ASH (Djurisic et al., 2023), ReAct (Sun et al., 2021), SCALE (Xu et al., 2024), and ReweightOOD (Regmi et al., 2024). †: Alternative training OOD method, *: Accuracy value inherited from ResNet18's classifier.

| Model | Pre | Accuracy | Near | | | Far | | | | |
|---|---|---|---|---|---|---|---|---|---|---|
| | | | CIFAR 10 | Tiny Imagenet | Avg. | MNIST | Places365 | SVHN | Textures | Avg. |
| MSP | ResNet18 | **76.91**$_{(0.11)}$ | 78.66$_{(0.15)}$ | 81.98$_{(0.16)}$ | 80.32$_{(0.15)}$ | 75.76$_{(2.51)}$ | 79.16$_{(0.19)}$ | 79.20$_{(0.96)}$ | 77.62$_{(0.58)}$ | 77.94$_{(0.65)}$ |
| | W&R | 76.19$_{(0.04)}$ | 78.48$_{(0.03)}$ | 82.19$_{(0.18)}$ | 80.33$_{(0.10)}$ | 77.01$_{(2.10)}$ | 79.72$_{(0.09)}$ | 80.60$_{(1.21)}$ | 78.18$_{(0.40)}$ | 78.88$_{(0.35)}$ |
| Temp. MSP | ResNet18 | **76.91**$_{(0.11)}$ | **79.16**$_{(0.09)}$ | 82.25$_{(0.01)}$ | 80.71$_{(0.05)}$ | 77.94$_{(1.52)}$ | 78.69$_{(0.19)}$ | 81.10$_{(1.43)}$ | 78.07$_{(0.77)}$ | 78.95$_{(0.48)}$ |
| | W&R | 76.19$_{(0.04)}$ | 78.87$_{(0.02)}$ | 82.27$_{(0.16)}$ | 80.57$_{(0.07)}$ | 77.41$_{(1.95)}$ | 80.06$_{(0.05)}$ | 81.72$_{(0.80)}$ | 77.81$_{(0.40)}$ | 79.25$_{(0.32)}$ |
| MDS | ResNet18 | 76.10$_{(0.09)}$ | 55.87$_{(0.22)}$ | 61.84$_{(0.19)}$ | 58.86$_{(0.08)}$ | 67.47$_{(0.81)}$ | 63.18$_{(0.50)}$ | 70.24$_{(6.51)}$ | 76.26$_{(0.69)}$ | 69.29$_{(1.42)}$ |
| | W&R | 76.10$_{(0.09)}$ | 55.87$_{(0.22)}$ | 61.84$_{(0.19)}$ | 58.86$_{(0.08)}$ | 67.47$_{(0.81)}$ | 63.18$_{(0.50)}$ | 70.24$_{(6.51)}$ | 76.26$_{(0.69)}$ | 69.29$_{(1.42)}$ |
| RMDS | ResNet18 | 76.10$_{(0.09)}$ | 77.75$_{(0.19)}$ | 82.58$_{(0.02)}$ | 80.17$_{(0.09)}$ | 79.74$_{(2.49)}$ | 83.40$_{(0.46)}$ | 85.10$_{(1.06)}$ | 83.65$_{(0.52)}$ | 82.97$_{(0.42)}$ |
| | W&R | 76.10$_{(0.09)}$ | 77.75$_{(0.19)}$ | 82.58$_{(0.02)}$ | 80.17$_{(0.09)}$ | 79.74$_{(2.49)}$ | 83.40$_{(0.46)}$ | 85.10$_{(1.06)}$ | 83.65$_{(0.52)}$ | 82.97$_{(0.42)}$ |
| ASH | ResNet18 | 77.25* | — | — | 78.20 | — | — | — | — | 80.58 |
| ReAct | ResNet18 | 77.25* | — | — | 80.77 | — | — | — | — | 80.39 |
| SCALE | ResNet18 | 77.26* | — | — | **80.99** | — | — | — | — | 81.42 |
| ReweightOOD† | ResNet18 | — | — | — | 71.27 | — | — | — | — | **91.12** |
| | | | Hierarchical Gaussian DPMMs | | | | | | | |
| Tied | ResNet18 | 76.11$_{(0.10)}$ | 77.67$_{(0.19)}$ | 82.56$_{(0.02)}$ | 80.11$_{(0.09)}$ | 79.82$_{(2.47)}$ | 83.38$_{(0.47)}$ | **85.17**$_{(1.13)}$ | 83.88$_{(0.52)}$ | 83.07$_{(0.44)}$ |
| | W&R | 76.10$_{(0.09)}$ | 77.75$_{(0.19)}$ | 82.59$_{(0.02)}$ | 80.17$_{(0.09)}$ | 79.75$_{(2.49)}$ | **83.41**$_{(0.46)}$ | 85.11$_{(1.06)}$ | 83.65$_{(0.52)}$ | 82.98$_{(0.42)}$ |
| Full | ResNet18 | 76.79$_{(0.22)}$ | 76.81$_{(0.15)}$ | **82.97**$_{(0.15)}$ | 79.89$_{(0.05)}$ | 82.10$_{(1.98)}$ | 79.16$_{(0.13)}$ | 81.22$_{(2.34)}$ | 82.07$_{(0.61)}$ | 81.14$_{(0.19)}$ |
| | W&R | 76.64$_{(0.14)}$ | 76.04$_{(0.18)}$ | 82.40$_{(0.16)}$ | 79.22$_{(0.06)}$ | 82.10$_{(1.90)}$ | 78.82$_{(0.16)}$ | 81.20$_{(2.78)}$ | 82.66$_{(0.61)}$ | 81.20$_{(0.30)}$ |
| Diag. | ResNet18 | 75.54$_{(0.10)}$ | 74.31$_{(0.39)}$ | 81.50$_{(0.31)}$ | 77.91$_{(0.32)}$ | 79.12$_{(1.99)}$ | 78.93$_{(0.48)}$ | 82.20$_{(2.31)}$ | 84.02$_{(0.76)}$ | 81.07$_{(0.48)}$ |
| | W&R | 76.07$_{(0.19)}$ | 76.30$_{(0.28)}$ | 82.13$_{(0.04)}$ | 79.22$_{(0.13)}$ | 81.46$_{(2.08)}$ | 81.79$_{(0.48)}$ | 84.98$_{(1.65)}$ | 83.31$_{(0.54)}$ | 82.88$_{(0.59)}$ |
| Coupled Diag. | ResNet18 | 75.54$_{(0.08)}$ | 75.80$_{(0.35)}$ | 82.77$_{(0.25)}$ | 79.28$_{(0.28)}$ | 79.25$_{(1.79)}$ | 80.29$_{(0.39)}$ | 83.29$_{(1.63)}$ | **84.77**$_{(0.67)}$ | 81.90$_{(0.43)}$ |
| | W&R | 76.04$_{(0.25)}$ | 76.18$_{(0.10)}$ | 79.93$_{(0.33)}$ | 78.05$_{(0.14)}$ | 75.97$_{(2.92)}$ | 79.80$_{(0.23)}$ | 80.58$_{(0.05)}$ | 80.81$_{(0.21)}$ | 79.29$_{(0.74)}$ |

Table 7: Performance of the Hierarchical Gaussian DPMM and baseline methods on the OpenOOD CIFAR-100 benchmark.

| Model | Accuracy | Near | | | Far | | | | |
|---|---|---|---|---|---|---|---|---|---|
| | | CIFAR 10 | Tiny Imagenet | Avg. | MNIST | Places365 | SVHN | Textures | Avg. |
| MSP | 76.19 | 78.48 | 82.19 | 80.33 | 77.01 | 79.72 | 80.60 | 78.18 | 78.88 |
| Temp. MSP | 76.19 | **78.87** | 82.27 | **80.57** | 77.41 | 80.06 | 81.72 | 77.81 | 79.25 |
| MDS | 76.10 | 55.87 | 61.84 | 58.86 | 67.47 | 63.18 | 70.24 | 76.26 | 69.29 |
| RMDS | 76.10 | 77.75 | 82.58 | 80.17 | 79.74 | 83.40 | 85.10 | **83.65** | 82.97 |
| | | Hierarchical Gaussian DPMMs | | | | | | | |
| Tied | 76.10 | 77.75 | **82.59** | 80.17 | 79.75 | **83.41** | 85.11 | 83.65 | **82.98** |
| Full | **76.64** | 76.04 | 82.40 | 79.22 | 82.10 | 78.82 | 81.20 | 82.66 | 81.20 |
| Diag. | 76.07 | 76.30 | 82.13 | 79.22 | 81.46 | 81.79 | 84.98 | 83.31 | 82.88 |
| Coupled Diag. | 76.04 | 76.18 | 79.93 | 78.05 | 75.97 | 79.80 | 80.58 | 80.81 | 79.29 |

## M    Feature Dimension Study with Independent RMDS

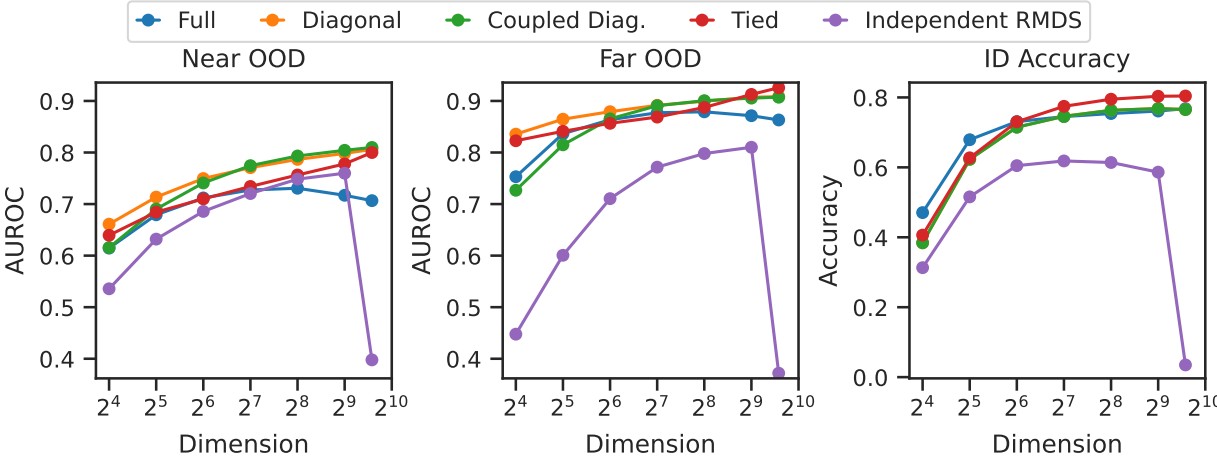

Figure 7: Performance on "near OOD", "far OOD", and in-distribution classification as a function of the feature dimension on the Imagenet-1K task. We projected the 768-dimensional ViT-B-16 features into lower dimensions using PCA, then projected into the eigenspace of the average within-class covariance. We compared the tied model (with full covariance) to the hierarchical model with full, diagonal, and coupled diagonal covariance and measured performance by area under the receiver operator curve (AUROC).

