# OpenReview forum: "A Bayesian Nonparametric Perspective on Mahalanobis Distance for Out of Distribution Detection"
_TMLR — Accepted by TMLR_

### Review · Reviewer_pzYS · 2025-09-08

**Summary Of Contributions:**

Strengths

1.Rigorous proofs (Propositions 4.1–4.2) formalize the RMDS-DPMM connection, filling a gap between heuristic distance-based methods and BNP generative modeling.

2.Hierarchical models address real-world limitations (e.g., heterogeneous covariances in ImageNet-1K, verified via Förstner-Moonen distance analysis) and are computationally feasible (EM runs in minutes, even for high-dimensional embeddings).

3.Tests on synthetic data (to isolate covariance/sample size effects) and OpenOOD (to validate real-world performance) with ablation studies (e.g., embedding dimensionality, preprocessing) ensure results are robust.

Weaknesses

1.The coupled diagonal model exhibits significant performance variation across datasets. On ImageNet-1K, it performs optimally in Near OOD detection, achieving an AUROC of 80.98—outperforming the tied model (79.28) and full covariance model (70.66) . However, on CIFAR-10 and CIFAR-100, its performance deteriorates markedly. For CIFAR-10, the coupled diagonal model’s Near OOD AUROC is 88.30, which is lower than the full covariance model’s 90.63 and the tied/RMDS models’ 89.83 . For CIFAR-100, its Near OOD AUROC drops to 78.05, trailing the tied model (80.17) and RMDS (80.17) . The paper fails to provide a clear explanation for this dataset-dependent behavior: it does not analyze how differences between datasets (e.g., ImageNet-1K’s 1000 classes with heterogeneous covariance vs. CIFAR-10’s 10 classes with more homogeneous covariance, or ViT vs. ResNet embeddings) interact with the coupled diagonal model’s design (e.g., its $\gamma_k$ scale factor for cross-dimensional variance correlation) to cause performance gaps.

2. The paper only conducts ViT-B for ImageNet-1k and ResNet-18 for CIFAR. How about try one or two more different models on these dataset.

3.The paper does not compare with state-of-the-art OOD methods from 2024 or non-Gaussian BNP models, which may limit context for its advantages. The baselines are too limited and old, please compare with more latest methods [1-5].

[1] Djurisic, Andrija, et al. "Extremely simple activation shaping for out-of-distribution detection." arXiv preprint arXiv:2209.09858 (2022).

[2] Jiang, Xue, et al. "Detecting out-of-distribution data through in-distribution class prior." International conference on machine learning. PMLR, 2023.

[3] Sun, Yiyou, Chuan Guo, and Yixuan Li. "React: Out-of-distribution detection with rectified activations." Advances in neural information processing systems 34 (2021): 144-157.

[4]  Xu, Kai, et al. "Scaling for training time and post-hoc out-of-distribution detection enhancement." arXiv preprint arXiv:2310.00227 (2023).

[5] Regmi, Sudarshan, et al. "Reweightood: Loss reweighting for distance-based ood detection." Proceedings of the IEEE/CVF Conference on Computer Vision and Pattern Recognition. 2024.

**Audience:**

Yes

**Audience Explanation:**

The paper revitalizes BNP methods for OOD detection, a task where BNP’s strength in handling infinite clusters (novel classes) has been underutilized. The DPMM-RMDS connection bridges BNP theory to practical, widely used distance-based methods, opening new directions for BNP applications in anomaly detection.

**Claims And Evidence:**

Yes

**Claims Explanation:**

1.Propositions 4.1 and 4.2 are rigorously proven (with full details in Appendix D) to link RMDS to Gaussian DPMM inlier probabilities. For example, Proposition 4.2 bounds the difference between DPMM scores and RMDS under sub-Gaussian data, using tools like Hanson-Wright inequality and Loewner-Heinz inequality to ensure correctness.

2.Synthetic experiments (Figure 3) isolate key variables (covariance heterogeneity via \(\nu_0\), per-class sample size \(N_k\)) and show: (a) hierarchical DPMMs outperform RMDS when \(\nu_0\) is small (heterogeneous covariances), (b) DPMMs dominate "Independent RMDS" when \(N_k/D\) is small (small samples vs. dimensions). These results directly validate the model’s design goals.

3.On OpenOOD (Table 1, Appendices I–K), hierarchical models outperform baselines in targeted regimes: (a) full covariance model achieves highest AUROC (93.50) on CIFAR-10 Far OOD, (b) coupled diagonal model leads on ImageNet-1K Near OOD (80.98 AUROC). Preprocessing ablation (e.g., W&R vs. PCA) and dimensionality analysis (Figure 4) confirm results are not artifacts of data preparation.

**Requested Changes:**

Please refer to weakness.

---

> ### Author Response · Authors · 2026-01-14
>
> > 1. The coupled diagonal model exhibits significant performance variation across datasets. On ImageNet-1K, it performs optimally in Near OOD detection, achieving an AUROC of 80.98—outperforming the tied model (79.28) and full covariance model (70.66) . However, on CIFAR-10 and CIFAR-100, its performance deteriorates markedly. For CIFAR-10, the coupled diagonal model’s Near OOD AUROC is 88.30, which is lower than the full covariance model’s 90.63 and the tied/RMDS models’ 89.83 . For CIFAR-100, its Near OOD AUROC drops to 78.05, trailing the tied model (80.17) and RMDS (80.17) . The paper fails to provide a clear explanation for this dataset-dependent behavior: it does not analyze how differences between datasets (e.g., ImageNet-1K’s 1000 classes with heterogeneous covariance vs. CIFAR-10’s 10 classes with more homogeneous covariance, or ViT vs. ResNet embeddings) interact with the coupled diagonal model’s design (e.g., its $\gamma_k$ scale factor for cross-dimensional variance correlation) to cause performance gaps.
>
> This observation is very important to understanding the model’s performance under different data distributions. Reviewer xdPP had a similar concern and we refer you to our response to xdPP, specifically the third bullet paragraph 2. Briefly, a theme of this work is the trade-off between the validity of the assumptions our models make and the resiliency to computational and statistical challenges. The coupled diagonal covariance model was designed to fit the observed Imagenet ViT features while remaining simple enough to fit the model’s parameters efficiently. This explains the relatively improved performance on Imagenet. As you noted the full covariance model is effective on CIFAR-10. This explains how the more expressive models excel when the feature dimension and distribution allow for good parameter estimates. Please refer to the 4th bullet 2nd paragraph of our response to Reviewer xdPP for further description of the CIFAR-10 performance.
>
> > 2. The paper only conducts ViT-B for ImageNet-1k and ResNet-18 for CIFAR. How about try one or two more different models on these dataset.
>
>  This is another great observation that is shared with Reviewer xdPP.
>
> The choice of feature extractor is an important design choice for these models. In particular, models that produce very high dimensional feature spaces with many degenerate or redundant dimensions pose statistical modeling challenges. In fact, this is one reason we explore ViTs for Imagenet, dimension 768, compared to ResNet50 with 2048 dimensional features. The higher dimension ResNet50 embeddings exacerbate the numerical stability and dimensional collapse problems of the smaller ViT embeddings especially for methods using full rank covariance matrices (RMDS, MDS, Tied DPMM and Full DPMM). The RMDS / MDS performance degrades when moving to ResNet50 features. In fact, RMDS methods are the top performing methods with ViT features while they don’t crack the top 10 on ResNet50 features according to the OpenOOD benchmark’s published performance numbers. For this reason, we felt that performing our Imagenet experiments on ResNet50 features was not worth the computational cost. We expect these factors to cause our models to suffer a similar drop in performance as the RMDS models.
>
> Notably, the CIFAR experiments utilize ResNet18 features with a dimension of 512. The competitive performance of our methods on the ResNet18 features compared to non-RMDS baselines show that the convolutional network architecture is not the cause for performance degradation of the Imagenet RMDS methods on ResNet50 features. Finally, we observe the state-of-the-art performance of the full covariance DPMM on CIFAR-10. The smaller feature dimension (512), lower number of classes (10), and high number of samples per class (5000) allow for better covariance estimation and the full covariance DPMM to outperform.
>
> > 3. The paper does not compare with state-of-the-art OOD methods from 2024 or non-Gaussian BNP models, which may limit context for its advantages. The baselines are too limited and old, please compare with more latest methods [1-5].
>
> Please refer to the updated tables 2, 4, and 6. We have added the available performance numbers for these methods according to the OpenOOD benchmark’s reported numbers.

---

### Review · Reviewer_xay4 · 2025-10-01

**Summary Of Contributions:**

The paper establishes a theoretical link between the Relative Mahalanobis Distance Score (RMDS) and Gaussian Dirichlet Process Mixture Models (DPMMs) with tied covariance. Building on this, it proposes hierarchical Gaussian DPMMs (full, diagonal, coupled diagonal) to model class-specific covariance structures.

**Additional Comments:**

Although post-hoc methods are the main focus of this paper, I think the authors still need to mention fine-tuning based methods in the related works. I listed below for some works that may of the practical interests.
[1]  Deep Anomaly Detection with Outlier Exposure
[2] POEM: Out-of-Distribution Detection with Posterior Sampling
[3] Learning to Augment Distributions for Out-of-Distribution Detection

It might be hard for the beginner to understand the basic problem setup of ood detection, considering add more detailed backgrounds about OOD detection in Sec 3. Also, the definition of \theta_k in Eq 2 seems to be missing, seemingly to be the joint of \mu and \simga. Similar issues also appear for the IW in Eq 12. I am also confused about why DPMM is of the particular interests of this paper, any particular benefits of it in the context of OOD detection, especially for the use of N_k and alpha in Eq. 1?

I am not sure if EM is indispensable for estimating v and kappa, is it possible to set them as hyperparameters? Because EM also cannot guarantee the convergence to the optimal solution.

If the proposed method still overfits to high dimensional data, I am quite concerned about its effectiveness. Two baseline methods may be required for comparisons, both in heuristics and experiments.
1. Estimating \mu and \sigma for each class directly by empirical expectation, further assuming that \sigma is diagonal.
2.  Conducting dimension reductions for to low dimensional space, maybe three to ten dimensions, then using empirical expectation to estimate \mu and \sigma

The paper is not easy to read, a lot of Bayesian designs are presented without detailed discussions.

More baselines and experimental setups are required for the experimental part.

**Audience:**

Yes

**Audience Explanation:**

OOD detection is an important area of machine learning, researchers in safety and trustworthiness may be interested in it.

**Broader Impact Concerns:**

Bayesian framework is of broad interest in the literature of OOD detection, the authors may contribute to this sub area.

**Claims And Evidence:**

No

**Claims Explanation:**

The authors pay a lot of effects in describe their algorithm, but realisation details and heuristics in their design choices are missing.

**Requested Changes:**

The clarity and the concreteness of this paper should be improved.

More experiments, baselines, and base models should be considered.

Please see Additional Comments for more comments.

---

> ### Author Response · Authors · 2026-01-14
>
> > Although post-hoc methods are the main focus of this paper, I think the authors still need to mention fine-tuning based methods in the related works. I listed below for some works that may of the practical interests. [1] Deep Anomaly Detection with Outlier Exposure [2] POEM: Out-of-Distribution Detection with Posterior Sampling [3] Learning to Augment Distributions for Out-of-Distribution Detection
>
> We thank the reviewer for highlighting these critical areas of OOD research. We agree that training time and outlier exposure methods are vital to the field. To provide readers with a comprehensive overview without deviating from the primary focus of this study, we have included citations to recent OOD detection surveys in the related works section.
>
> > It might be hard for the beginner to understand the basic problem setup of ood detection, considering add more detailed backgrounds about OOD detection in Sec 3. Also, the definition of $\theta_k$ in Eq 2 seems to be missing, seemingly to be the joint of $\mu$ and $\sigma$. Similar issues also appear for the IW in Eq 12. I am also confused about why DPMM is of the particular interests of this paper, any particular benefits of it in the context of OOD detection, especially for the use of $N_k$ and $\alpha$ in Eq. 1?
>
> We have added a brief description of the OOD detection task for image classification to the related works section. Readers are referred to prior OOD detection surveys for more detailed introduction to the problem.
>
> We clarify that $\theta_k$ is the set of cluster parameters by adding the notation to the sentence introducing Eq 2. Likewise, $\theta_k$ is defined in the sentence preceding Eq 12.
>
> > I am not sure if EM is indispensable for estimating v and kappa, is it possible to set them as hyperparameters? Because EM also cannot guarantee the convergence to the optimal solution.
>
> The EM algorithms we derive optimize the marginal likelihood, as you mentioned. In particular, EM can be used to find MAP (maximum a posteriori) solutions for hierarchical models with priors over the parameters like the full, diagonal, and coupled diagonal covariance hierarchical DPMMs we propose. We prefer EM over direct gradient descent on the marginal log likelihood because EM converges quickly with the closed-form updates available in these conjugate models, and it does not require setting learning rates or extra care to deal with constraints on the hyperparameters (e.g., some hyperparameters must be positive definite matrices). That said, variants of gradient descent would be fine alternatives.
>
> Finally, as the reviewer suggests, we could manually tune the $\nu$ and $\kappa$ hyperparameters. The downsides of this approach is that it adds more hyperparameters to the model. Each new dataset would require a hyperparameter search for these parameters which would be slower than the proposed EM solution.
>
> > If the proposed method still overfits to high dimensional data, I am quite concerned about its effectiveness. Two baseline methods may be required for comparisons, both in heuristics and experiments.
>
> Our experiments show that the proposed models are effective particularly on the Imagenet-1K dataset. We have added additional baselines as suggested by fellow reviewers. The proposed methods are some of the most effective strategies in the Imagenet-1K OpenOOD benchmark with ViT features.
>
> While we show that our method is effective and we believe that the claims are well supported as acknowledged by the other reviewers, we note that effectiveness (i.e., improvement over alternative approaches) is not a criteria for TMLR suitability (https://jmlr.org/tmlr/acceptance-criteria.html).
>
> > 1. Estimating $\mu$ and $\sigma$ for each class directly by empirical expectation, further assuming that $\sigma$ is diagonal.
>
> The independent RMDS model in our synthetic baselines is similar to what you suggest but with full covariances. We add Figure 7 to the appendix which is the dimension reduction study from the main text with the independent RMDS baseline. The performance of the independent RMDS baseline quickly degrades as the number of dimensions kept increases.
>
> > 2. Conducting dimension reductions for to low dimensional space, maybe three to ten dimensions, then using empirical expectation to estimate $\mu$ and $\sigma$
>
> Please refer to Figure 4 for our feature dimension reduction experiment.
>
> > The paper is not easy to read, a lot of Bayesian designs are presented without detailed discussions.
>
> We do assume familiarity with Bayesian models — particularly Bayesian nonparametric mixture models — and a full introduction to this subject would be beyond the scope of the paper. However, we have added further references to textbook treatments of this subject for a broader audience that may not be familiar with this background material.

---

### Review · Reviewer_xdPP · 2025-12-30

**Summary Of Contributions:**

Theoretically, this paper establishes a bridge between the Relative Mahalanobis Distance Score (RMDS) and the Dirichlet Process Mixture Model (DPMM). Moreover, the authors show that the DPMM score from a Gaussian DPMM with tied covariance may correlate perfectly with the RMDS. In addition, three hierarchical Gaussian DPMMs (Full, Diagonal, and Coupled Diagonal) are proposed, which generalize the RMDS.

From the experimental perspective, the authors analyze the results and differences of these Gaussian DPMMs (Full, Diagonal, and Coupled Diagonal) on two-dimensional synthetic data. Furthermore, they conduct experiments on the OpenOOD benchmark. By evaluating the performance of different models on embeddings from CIFAR-10 and CIFAR-100 using ResNet-18, as well as embeddings from ImageNet-1K using ViT-B/16, they draw the conclusion that Bayesian nonparametric methods achieve superior performance when covariance structures differ, or training data is scarce.

**Audience:**

Yes

**Audience Explanation:**

This paper addresses the problem of out-of-distribution (OOD) detection, which is a popular and rapidly growing topic in the field of machine learning.

**Claims And Evidence:**

Yes

**Claims Explanation:**

The authors provide theoretical derivations, along with visualizations of the synthesized data and numerical results on CIFAR10/100 and ImageNet-1k.

**Requested Changes:**

- I would recommend that the authors include a concrete **algorithm flowchart or table** after presenting the formulas for the three models in the Methods section. This would help to clearly illustrate processes such as PCA and EM, improving readability. Currently, most of the PCA and EM operations are described in the appendix; it is suggested to briefly summarize these procedures in the main text.

- The authors may include an analysis of the **computational complexity** of the baseline method RMDS, and the three Hierarchical Gaussian DPMM methods proposed in this paper, or to report the actual runtime of experiments. This would demonstrate that the Hierarchical Gaussian DPMMs are not significantly more complex than RMDS, thereby supporting their practical applicability.

- In Section 6.1, the paper conducts experiments using two-dimensional **synthetic data**. This raises several questions regarding experiments on real data: Are the results on real data similar to those on synthetic data? Admittedly, reducing high-dimensional complex image data (e.g., CIFAR, ImageNet) to two dimensions results in significant information loss, but what happens with simpler image datasets, such as MNIST? If the results on real data are similar to those on synthetic data, what exactly are those results? Conversely, if the results on real data differ substantially from those on synthetic data, what justifies using synthetic data for experiments? What causes these differences, and how might they affect the conclusions drawn in this paper?

- How does the choice of **feature extractor** (i.e., different foundation models) affect the experimental results? For instance, this paper observes relatively poor performance on ImageNet (using ViT-B/16), which may be due to the high-dimensional embeddings. How would the results change if embeddings of lower dimensionality were obtained using other models, such as ResNet?

- Since the differences in classification results are generally small, including the standard deviation (std) of the results could better reflect the stability of the models, making the findings more convincing—at least in the main figures and tables, such as Table 1.

---

> ### Author Response · Authors · 2026-01-14
>
> > I would recommend that the authors include a concrete algorithm flowchart or table after presenting the formulas for the three models in the Methods section. This would help to clearly illustrate processes such as PCA and EM, improving readability. Currently, most of the PCA and EM operations are described in the appendix; it is suggested to briefly summarize these procedures in the main text.
>
> Thank you for this suggestion. We have added Figure 5 to illustrate the steps of the preprocessing pipeline to map images to features x for OOD detection.
>
> > The authors may include an analysis of the computational complexity of the baseline method RMDS, and the three Hierarchical Gaussian DPMM methods proposed in this paper, or to report the actual runtime of experiments. This would demonstrate that the Hierarchical Gaussian DPMMs are not significantly more complex than RMDS, thereby supporting their practical applicability.
>
> Please see the added discussion of computational complexity to Appendix D. The DPMM based approaches add some space and time complexity, as expected since the models are more expressive with per-class covariances, but the costs are quite reasonable in practice.
>
> > In Section 6.1, the paper conducts experiments using two-dimensional synthetic data. This raises several questions regarding experiments on real data: Are the results on real data similar to those on synthetic data? Admittedly, reducing high-dimensional complex image data (e.g., CIFAR, ImageNet) to two dimensions results in significant information loss, but what happens with simpler image datasets, such as MNIST? If the results on real data are similar to those on synthetic data, what exactly are those results? Conversely, if the results on real data differ substantially from those on synthetic data, what justifies using synthetic data for experiments? What causes these differences, and how might they affect the conclusions drawn in this paper?
>
> The goal of the synthetic experiments was to understand the performance of each model under different data distributions in a controllable manner. By doing so we can tease out the sensitivity of each method when their assumptions are invalidated to small or large degrees. For example, we show that models with a shared covariance assumption (i.e. tied DPMM and RMDS) are sensitive to datasets where that assumption is invalid (e.g. $\nu_0=4$). However, the diagonal and full covariance DPMMs are resilient as their assumptions still hold. Likewise, when we consider the other extreme, the independent covariance RMDS, the model requires a large number of samples to approximate the class covariances evidenced by the poor performance when $N_k=5$. This type of experimentation provides valuable information which is not feasible to gather with real image datasets.
>
> Synthetic experiments do not pose the same statistical estimation challenges provided by high dimensional image features from a deep network. Indeed, high dimensional features spaces pose several challenges, such as dimensional collapse and under- or over-fitting class covariances, which are a main theme of the paper. Our experimental results reflect this reality as we see that lower capacity methods such as RMDS and the tied DPMM perform relatively well (see Table 1) despite their misspecification (see Figure 1). We propose methods that attempt to strike the right balance between assumption validity and resiliency to computational and statistical challenges. Specifically the coupled diagonal covariance model was designed to fit the observed Imagenet ViT features while remaining simple enough to fit efficiently.

---

> > ### Author Response · Authors · 2026-01-14
> >
> > > How does the choice of feature extractor (i.e., different foundation models) affect the experimental results? For instance, this paper observes relatively poor performance on ImageNet (using ViT-B/16), which may be due to the high-dimensional embeddings. How would the results change if embeddings of lower dimensionality were obtained using other models, such as ResNet?
> >
> > The choice of feature extractor is an important design choice for these models. In particular, models that produce very high dimensional feature spaces with many degenerate or redundant dimensions pose statistical modeling challenges. In fact, this is one reason we explore ViTs for Imagenet, dimension 768, compared to ResNet50 with 2048 dimensional features. The higher dimension ResNet50 embeddings exacerbate the numerical stability and dimensional collapse problems of the smaller ViT embeddings especially for methods using full rank covariance matrices (RMDS, MDS, Tied DPMM and Full DPMM). The RMDS / MDS performance degrades when moving to ResNet50 features. In fact, RMDS methods are the top performing methods with ViT features while they don’t crack the top 10 on ResNet50 features according to the OpenOOD benchmark’s published performance numbers. For this reason, we felt that performing our Imagenet experiments on ResNet50 features was not worth the computational cost. We expect these factors to cause our models to suffer a similar drop in performance as the RMDS models.
> >
> > Notably, the CIFAR experiments utilize ResNet18 features with a dimension of 512. The competitive performance of our methods on the ResNet18 features compared to non-RMDS baselines show that the convolutional network architecture is not the cause for performance degradation of the Imagenet RMDS methods on ResNet50 features. Finally, we observe the state-of-the-art performance of the full covariance DPMM on CIFAR-10. The smaller feature dimension (512), lower number of classes (10), and high number of samples per class (5000) allow for better covariance estimation and the full covariance DPMM to outperform.
> >
> > > Since the differences in classification results are generally small, including the standard deviation (std) of the results could better reflect the stability of the models, making the findings more convincing—at least in the main figures and tables, such as Table 1.
> >
> > Thank you for the suggestion. We are only able to provide a single score for the Imagenet experiments as the OpenOOD benchmark only provides 1 set of ViT weights. The ViT features and the proposed methods are deterministic given these weights.
> >
> > We provide the CIFAR-10/-100 standard deviations in the updated tables. Three sets of ResNet18 weights trained with different random seeds are available from the OpenOOD benchmark. The standard deviation across seeds is small compared to the value as seen in Tables 4 and 6. Note we did not place the standard deviations in the main text to avoid cluttering Table 1.

---

### Author Response · Authors · 2026-01-14

We thank the reviewers for their positive reception and constructive feedback. We respond to each of the reviewers individually. We hope that our responses address the reviewers’ concerns.

---

### Decision · Action_Editor_tYxB · 2026-01-31

**Recommendation:** Accept with minor revision

**Audience:**

Yes

**Audience Explanation:**

OOD detection is an important problem in the machine learning field. TMLR's audience will be interested in knowing the findings of this paper.

**Claims And Evidence:**

Yes

**Claims Explanation:**

This paper made a contribution to the OOD detection field with both theoretical and empirical results. All claims are supported by the evidence (e.g., theorems and empirical experiments). However, the authors could further enhance the clarity of this paper, as per the Reviewer xdPP's comments.

---

> ### Author Response · Authors · 2026-02-23
>
> We thank the action editor for leading the review process and recommending our work for acceptance at TMLR.  We have addressed the reviewers’ concerns in our camera ready version. We list the updates below.
> - We have added Figure 5 to illustrate the steps of the preprocessing pipeline as recommended by reviewer xdPP.
> - We also add a section in the appendix that discusses the computational complexity of our method.
> - For the CIFAR experiments we report the standard deviations on the performance metrics.
> - We have added clarifications and pointers to background on OOD detection and Bayesian modeling as requested by reviewer xay4
> - We have reported additional baselines to our results for methods with previously published performance metrics on the OpenOOD baseline as requested by reviewer pzYS
>
> We appreciate the reviewers' suggestions. They have helped strengthen the paper.